# SKIMMR: facilitating knowledge discovery in life sciences by machine-aided skim reading

Vít Nováček[1] and Gully A.P.C. Burns[2]

[1] Insight Centre (formerly DERI), National University of Ireland Galway, Galway, Ireland
[2] Information Sciences Institute, University of Southern California, Marina del Rey, CA, USA

Corresponding author
Vít Nováček, vit.novacek@deri.org

## ABSTRACT

**Background.** Unlike full reading, 'skim-reading' involves the process of looking quickly over information in an attempt to cover more material whilst still being able to retain a superficial view of the underlying content. Within this work, we specifically emulate this natural human activity by providing a dynamic graph-based view of entities automatically extracted from text. For the extraction, we use shallow parsing, co-occurrence analysis and semantic similarity computation techniques. Our main motivation is to assist biomedical researchers and clinicians in coping with increasingly large amounts of potentially relevant articles that are being published ongoingly in life sciences.

**Methods.** To construct the high-level network overview of articles, we extract weighted binary statements from the text. We consider two types of these statements, co-occurrence and similarity, both organised in the same distributional representation (i.e., in a vector-space model). For the co-occurrence weights, we use point-wise mutual information that indicates the degree of non-random association between two co-occurring entities. For computing the similarity statement weights, we use cosine distance based on the relevant co-occurrence vectors. These statements are used to build fuzzy indices of terms, statements and provenance article identifiers, which support fuzzy querying and subsequent result ranking. These indexing and querying processes are then used to construct a graph-based interface for searching and browsing entity networks extracted from articles, as well as articles relevant to the networks being browsed. Last but not least, we describe a methodology for automated experimental evaluation of the presented approach. The method uses formal comparison of the graphs generated by our tool to relevant gold standards based on manually curated PubMed, TREC challenge and MeSH data.

**Results.** We provide a web-based prototype (called 'SKIMMR') that generates a network of inter-related entities from a set of documents which a user may explore through our interface. When a particular area of the entity network looks interesting to a user, the tool displays the documents that are the most relevant to those entities of interest currently shown in the network. We present this as a methodology for browsing a collection of research articles. To illustrate the practical applicability of SKIMMR, we present examples of its use in the domains of Spinal Muscular Atrophy and Parkinson's Disease. Finally, we report on the results of experimental evaluation using the two domains and one additional dataset based on the TREC challenge. The results show how the presented method for machine-aided skim reading

**Peer**J ________________________________________

outperforms tools like PubMed regarding focused browsing and informativeness of the browsing context.

## INTRODUCTION

In recent years, knowledge workers in life sciences are increasingly overwhelmed by an ever-growing quantity of information. PubMed[1] contained more than 23 million abstracts as of November 2013, with a new entry being added every minute. The current textual content available online as PubMed abstracts amount to over 2 billion words (based on estimates derived from a random sample of about 7,000 records). Information retrieval technology helps researchers pinpoint individual papers of interest within the overall mass of documents, but how can scientists use that to acquire a sense of the overall organization of the field? How can users discover new knowledge within the literature when they might not know what they are looking for ahead of time?

Strategic reading aided by computerised solutions may soon become essential for scientists (*Renear & Palmer, 2009*). Our goal is to provide a system that can assist readers to explore large numbers of documents efficiently. We present 'machine-aided skim-reading' as a way to extend the traditional paradigm of searching and browsing a text collection (in this case, PubMed abstracts) through the use of a search tool. Instead of issuing a series of queries to reveal lists of ranked documents that may contain elements of interest, we let the user search and browse a *network of entities and relations* that are explicitly or implicitly present in the texts. This provides a simplified and high-level overview of the domain covered by the text, and allows users to identify and focus on items of interest without having to read any text directly. Upon discovering an entity of interest, the user may transition from our 'skimming' approach to read the relevant texts as needed.

This article is organised as follows. 'Methods' describes methods used in SKIMMR for: (1) extraction of biomedical entities from data; (2) computation of the co-occurrence and similarity relationships between the entities; (3) indexing and querying of the resulting knowledge base; (4) evaluating the knowledge base using automated simulations. Each of the methods is explained using examples. 'Results' presents the SKIMMR prototype and explains typical usage of the tool in examples based on user interactions. We also describe evaluation experiments performed with three different instances of the tool. In 'Discussion' we discuss the results, give an overview of related work and outline our future directions. There is also 'Formulae Definitions' that provides details on some of the more complex formulae introduced in the main text.

The main contributions of the presented work are: (1) machine-aided skim-reading as a new approach to semi-automated knowledge discovery; (2) fuzzy indexing and querying method for efficient on-demand construction and presentation of the high-level

[1] The central US repository of published papers in the life sciences since the 1950s, see http://www.ncbi.nlm.nih.gov/pubmed.
graph-based article summaries; (3) detailed examples that explain the applied methods in a step-by-step fashion even to people with little or no computer science background; (4) an open-source prototype implementing the described method, readily available for processing custom data, and also in the form of two pre-computed instances deployed on Spinal Muscular Atrophy and Parkinson's Disease data; (5) an evaluation methodology based on simulations and formally defined measures of semantic coherence, information content and complexity that can be used not only for evaluating SKIMMR (as we did in the article), but also for assessment of other tools and data sets utilising graph structures.

## METHODS

This section describes how the knowledge base supporting the process of machine-aided skim reading is generated from the input data (i.e., biomedical articles and data). Firstly we describe extraction of entities and basic co-occurrence relationships between them ('Extracting basic co-occurrence statements from texts'). 'Computing a knowledge base from the extracted statements' is about how we compute more general, corpus-wide relationships from the basic extracted co-occurrence statements. 'Indexing and querying the knowledge base' explains how the processed content can be indexed and queried in order to generate the graph-based summaries with links to the original documents. Finally, 'Evaluation methodology' introduces a method for a simulation-based evaluation of the generated content in the context of machine-aided skim reading. For the research reported in this article, we received an exemption from IRB review by the USC UPIRB, under approval number UP-12-00414.

### Extracting basic co-occurrence statements from texts

We process the abstracts by a biomedical text-mining tool[2] in order to extract named entities (e.g., drugs, genes, diseases or cells) from the text. For each abstract with a PubMed ID *PMID*, we produce a set of $(e_x, e_y, cooc((e_x, e_y), PubMed_{PMID}), PubMed_{PMID})$ tuples, where $e_x, e_y$ range over all pairs of named entities in the abstract with the *PMID* identifier, and $cooc((e_x, e_y), PubMed_{PMID})$ is a co-occurrence score of the two entities computed using the formula (1) detailed in 'Co-occurrences'. The computation of the score is illustrated in the following example.

**Example 1** *Imagine we want to investigate the co-occurrence of the* `parkinsonism` *and* DRD *(dopamine-responsive dystopia) concepts in a data set of PubMed abstracts concerned with clinical aspects of Parkinson's disease.*[3] *There are two articles in the data set where the corresponding terms co-occur:*

- *Jeon BS, et al.* Dopamine transporter density measured by 123Ibeta-CIT single-photon emission computed tomography is normal in dopa-responsive dystonia *(PubMed ID: 9629849).*
- *Snow BJ, et al.* Positron emission tomographic studies of dopa-responsive dystonia and early-onset idiopathic parkinsonism *(PubMed ID: 8239569).*

[2] A part of the LingPipe suite, see http://alias-i.com/lingpipe/ for details.

[3] Which we have processed in one of the pre-computed instances of SKIMMR, see 'Parkinson's disease' for details.

*The relevant portions of the first abstract (PubMed ID: 9629849) are summarised in the following table (split into sentences numbered from the beginning of the text):*

| ... | ... |
|-----|-----|
| 12 | Therefore, we performed 123Ibeta-CIT single-photon emission computed tomography (123Ibeta-CIT SPECT) in clinically diagnosed DRD, PD, and JPD, and examined whether DAT imaging can differentiate DRD from PD and JPD. |
| ... | ... |
| 14 | Five females (4 from two families, and 1 sporadic) were diagnosed as DRD based on early-onset foot dystonia and progressive parkinsonism beginning at ages 7–12. |
| ... | ... |
| 17 | 123Ibeta-CIT striatal binding was normal in DRD, whereas it was markedly decreased in PD and JPD. |
| ... | ... |
| 22 | A normal striatal DAT in a parkinsonian patient is evidence for a nondegenerative cause of parkinsonism and differentiates DRD from JPD. |
| 23 | Finding a new mutation in one family and failure to demonstrate mutations in the putative gene in other cases supports the usefulness of DAT imaging in diagnosing DRD. |

*Based on the sentence numbers in the excerpt, we can compute the co-occurrence score of the* (parkinsonism, DRD) *tuple as:*

$$cooc((\text{parkinsonism, DRD}), PubMed_{9629849}) = \left(1 + \frac{1}{4} + \frac{1}{3} + \frac{1}{3}\right) + \left(1 + \frac{1}{2}\right) = 3.41\bar{6}.$$

*Similar to the above, the portions relevant to the* (parkinsonism, DRD) *co-occurrences according to the second abstract (PubMed ID: 8239569) are as follows:*

| 1 | There are two major syndromes presenting in the early decades of life with dystonia and parkinsonism: dopa-responsive dystonia (DRD) and early-onset idiopathic parkinsonism (EOIP). |
|---|---|
| 2 | DRD presents predominantly in childhood with prominent dystonia and lesser degrees of parkinsonism. |
| ... | ... |
| 5 | Some have suggested, however, that DRD is a form of EOIP. |
| ... | ... |

*The co-occurrence score is then:*

$$cooc((\text{parkinsonism, DRD}), PubMed_{8239569}) = \left(1 + \frac{1}{2} + 1 + \frac{1}{2}\right) + \frac{1}{4} = 3.25.$$

*Therefore the basic co-occurrence tuples produced from the two articles are:*

$$(\text{parkinsonism, DRD}, 3.41\bar{6}, PubMed_{9629849}),$$

$$(\text{parkinsonism, DRD}, 3.25, PubMed_{8239569}).$$

## Computing a knowledge base from the extracted statements

From the basic co-occurrence statements, we compute a knowledge base, which is a comprehensive network of interlinked entities. This network supports the process of navigating a skeletal structure of the knowledge represented by the corpus of the input PubMed articles (i.e., the actual skim reading). The knowledge base consists of two types of statements: (1) corpus-wide co-occurrence and (2) similarity. The way to compute the particular types of statements in the knowledge base is described in the following two sections.

### *Corpus-wide co-occurrence*

The basic co-occurrence tuples extracted from the PubMed abstracts only express the co-occurrence scores at the level of particular documents. We need to aggregate these scores to examine co-occurrence across the whole corpus. For that, we use point-wise mutual information (*Manning, Raghavan & Schütze, 2008*), which determines how much two co-occurring terms are associated or disassociated, comparing their joint and individual distributions over a data set. We multiply the point-wise mutual information value by the absolute frequency of the co-occurrence in the corpus to prioritise more frequent phenomena. Finally, we filter and normalise values so that the results contain only scores in the [0, 1] range. The scores are computed using the formulae (2)–(5) in 'Co-occurrences'.

The aggregated co-occurrence statements that are added to the knowledge base are in the form of $(x, cooc, y, \nu(fpmi(x, y), P))$ triples, where $x, y$ range through all terms in the basic co-occurrence statements, the scores are computed across all the documents where $x, y$ co-occur, and the *cooc* expression indicates co-occurrence as the actual type of the relation between $x, y$. Note that the co-occurrence relation is symmetric, meaning that if $(x, cooc, y, w_1)$ and $(y, cooc, x, w_2)$ are in the knowledge base, $w_1$ must be equal to $w_2$.

**Example 2** *Assuming our corpus consists only of the two articles from Example 1, the point-wise mutual information score of the* (parkinsonism, DRD) *tuple can be computed using the following data:*

- $p$(parkinsonism, DRD)*–joint distribution of the* (parkinsonism, DRD) *tuple within all the tuples extracted from the PubMed abstracts with IDs 9629849 and 8239569, which equals* $3.41\bar{6} + 3.25 = 6.\bar{6}$ *(sum across all the* (parkinsonism, DRD) *basic co-occurrence tuples);*
- $p$(parkinsonism), $p$(DRD)*–individual distributions of the* parkinsonism, DRD *arguments within all extracted tuples, which equal 28.987 and 220.354, respectively (sums of the weights in all basic co-occurrence statements that contain* parkinsonism *or* DRD *as one of the arguments, respectively);*
- $F$(parkinsonism, DRD), $|T|$*–the absolute frequency of the* parkinsonism, DRD *co-occurrence and the number of all basic co-occurrence statements extracted from the abstracts, which equals to 2 and 1,414, respectively;*
- $P$*–the percentile for the normalisation, equal to 95, which results in the normalisation constant 2.061 (a non-normalised score such that only 5% of the scores are higher than that).*

*The whole formula is then:*

$$npmi(\texttt{parkinsonism}, \texttt{DRD}) = \nu(fpmi(\texttt{parkinsonism}, \texttt{DRD}), P) =$$

$$= \nu(F(\texttt{parkinsonism}, \texttt{DRD}) \cdot log_2 \frac{p(\texttt{parkinsonism}, \texttt{DRD})}{p(\texttt{parkinsonism})p(\texttt{DRD})}, 95) \doteq$$

$$\doteq \frac{2 \cdot log_2 \frac{6.\bar{6}}{28.987 \cdot 220.354}}{2.061} \doteq 0.545.$$

*Thus the aggregated co-occurrence statement that is included in the knowledge base is*

$(\texttt{parkinsonism}, \texttt{cooc}, \texttt{DRD}, 0.545)$.

### Similarity

After having computed the aggregated and filtered co-occurrence statements, we add one more type of relationship–similarity. Many other authors have suggested ways for computing semantic similarity (see *d'Amato, 2007* for a comprehensive overview). We base our approach on cosine similarity, which has become one of the most commonly used approaches in information retrieval applications (*Singhal, 2001*; *Manning, Raghavan & Schütze, 2008*). The similarity and related notions are described in detail in 'Similarities', formulae (6) and (7).

Similarity indicates a higher-level type of relationship between entities that may not be covered by mere co-occurrence (entities not occurring in the same article may still be similar). This adds another perspective to the network of connections between entities extracted from literature, therefore it is useful to make similarity statements also a part of the SKIMMR knowledge base. To do so, we compute the similarity values between all combinations of entities $x, y$ and include the statements $(x, sim, y, sim(x, y))$ into the knowledge base whenever the similarity value is above a pre-defined threshold (0.25 is used in the current implementation).[4]

A worked example of how to compute similarity between two entities in the sample knowledge base is given below.

**Example 3** *Let us use* '$\texttt{parkinsonisms}$', '$\texttt{mrpi values}$' *as sample entities* $a, b$. *In a full version of Parkinson's disease knowledge base (that contains the data used in the previous examples, but also hundreds of thousands of other statements), there are* 19 *shared entities among the ones related to* $a, b$ *(for purposes of brevity, each item is linked to a short identifier to be used later on):* (1) $\texttt{msa} - \texttt{p} \sim t_0$, (2) $\texttt{clinically unclassifiable parkinsonism} \sim t_1$, (3) $\texttt{cup} \sim t_2$, (4) $\texttt{vertical ocular slowness} \sim t_3$, (5) $\texttt{baseline clinical evaluation} \sim t_4$, (6) $\texttt{mr} \sim t_5$, (7) $\texttt{parkinsonian disorders} \sim t_6$, (8) $\texttt{psp phenotypes} \sim t_7$, (9) $\texttt{duration} \sim t_8$, (10) $\texttt{patients} \sim t_9$, (11) $\texttt{clinical diagnostic criteria} \sim t_{10}$, (12) $\texttt{abnormal mrpi values} \sim t_{11}$, (13) $\texttt{pd} \sim t_{12}$, (14) $\texttt{magnetic resonance parkinsonism index} \sim t_{13}$, (15) $\texttt{parkinson disease} \sim t_{14}$, (16) $\texttt{mri} \sim t_{15}$, (17) $\texttt{parkinson's disease} \sim t_{16}$, (18) $\texttt{psp} \sim t_{17}$, (19) $\texttt{normal mrpi values} \sim t_{18}$.

[4] Similar to the co-occurrence statements described before, the *sim* expression refers to the type of the relation between $x, y$, i.e., similarity.

*The co-occurrence complements $a, b$ of the* `parkinsonisms`, `mrpi values` *entities (i.e., associated co-occurrence context vectors) are summarised in the following table:*

|   | $t_0$ | $t_1$ | $t_2$ | $t_3$ | $t_4$ | $t_5$ | $t_6$ | $t_7$ | $t_8$ | $t_{10}$ | $t_{11}$ | $t_{13}$ | $t_{14}$ | $t_{15}$ | $t_{17}$ | $t_{18}$ |
|---|---|---|---|---|---|---|---|---|---|---|---|---|---|---|---|---|
| $a$ | 0.14 | 0.39 | 1.0 | 0.08 | 0.26 | 0.06 | 0.18 | 0.4 | 0.07 | 0.27 | 0.09 | 0.7 | 0.03 | 0.14 | 0.33 | 0.25 |
| $b$ | 0.26 | 0.57 | 1.0 | 0.3 | 0.82 | 0.2 | 0.33 | 0.26 | 0.39 | 0.43 | 0.36 | 0.41 | 0.06 | 0.34 | 1.0 | 1.0 |

*Note that the elements $t_9, t_{12}, t_{16}$ are omitted since their weight in at least one of the complements is $<0.01$ and thus does not contribute significantly to the result. The sizes of the co-occurrence complement vectors are 3.048, 2.491 for* `parkinsonisms`, `mrpi values`, *respectively, while their dot product is 2.773. Therefore their similarity is equal to $\frac{2.773}{3.048 \cdot 2.491} \doteq 0.365$ and the new statement to be added to the knowledge base is*

(`parkinsonisms`, `sim`, `mrpi values`, 0.365).

## Indexing and querying the knowledge base

The main purpose of SKIMMR is to allow users to efficiently search and navigate in the SKIMMR knowledge bases, and retrieve articles related to the content discovered in the high-level entity networks. To support that, we maintain several indices of the knowledge base contents. The way how the indices are built and used in querying SKIMMR is described in the following two sections.

### Knowledge base indices

In order to expose the SKIMMR knowledge bases, we maintain three main indices: (1) a *term* index–a mapping from entity terms to other terms that are associated with them by a relationship (like co-occurrence or similarity); (2) a *statement* index–a mapping that determines which statements the particular terms occur in; (3) a *source* index–a mapping from statements to their sources, *i.e.*, the texts from which the statements have been computed. In addition to the main indices, we use a full-text index that maps spelling alternatives and synonyms to the terms in the term index.

The main indices are implemented as matrices that reflect the weights in the SKIMMR knowledge base:

|       | $T_1$ | $T_2$ | $\dots$ | $T_n$ |
|-------|-------|-------|---------|-------|
| $T_1$ | $t_{1,1}$ | $t_{1,2}$ | $\dots$ | $t_{1,n}$ |
| $T_2$ | $t_{2,1}$ | $t_{2,2}$ | $\dots$ | $t_{2,n}$ |
| $\vdots$ | $\vdots$ | $\vdots$ | $\ddots$ | $\vdots$ |
| $T_n$ | $t_{n,1}$ | $t_{n,2}$ | $\dots$ | $t_{n,n}$ |

|       | $S_1$ | $S_2$ | $\dots$ | $S_m$ |
|-------|-------|-------|---------|-------|
| $T_1$ | $s_{1,1}$ | $s_{1,2}$ | $\dots$ | $s_{1,m}$ |
| $T_2$ | $s_{2,1}$ | $s_{2,2}$ | $\dots$ | $s_{2,m}$ |
| $\vdots$ | $\vdots$ | $\vdots$ | $\ddots$ | $\vdots$ |
| $T_n$ | $s_{n,1}$ | $s_{n,2}$ | $\dots$ | $s_{n,m}$ |

|       | $P_1$ | $P_2$ | $\dots$ | $P_q$ |
|-------|-------|-------|---------|-------|
| $S_1$ | $p_{1,1}$ | $p_{1,2}$ | $\dots$ | $p_{1,q}$ |
| $S_2$ | $p_{2,1}$ | $p_{2,2}$ | $\dots$ | $p_{2,q}$ |
| $\vdots$ | $\vdots$ | $\vdots$ | $\ddots$ | $\vdots$ |
| $S_m$ | $p_{m,1}$ | $p_{m,2}$ | $\dots$ | $p_{m,q}$ |

where:

- $T_1, \dots, T_n$ are identifiers of all entity terms in the knowledge base and $t_{i,j} \in [0, 1]$ is the maximum weight among the statements of all types existing between entities $T_i, T_j$ in the knowledge base (0 if there is no such statement);

- $S_1, \ldots, S_m$ are identifiers of all statements present in the knowledge base and $s_{i,j} \in \{0, 1\}$ determines whether an entity $T_i$ occurs in a statement $S_j$ or not;
- $P_1, \ldots, P_q$ are identifiers of all input textual resources, and $p_{i,j} \in [0, 1]$ is the weight of the statement $S_i$ if $P_j$ was used in order to compute it, or zero otherwise.

**Example 4** *To illustrate the notion of the knowledge base indices, let us consider a simple knowledge base with only two statements from Examples 1 and 3: $S_1 \sim$ (parkinsonism, cooc, DRD, 0.545), $S_2 \sim$ (parkinsonisms, sim, mrpi values, 0.365). Furthermore, let us assume that: (i) the statement $S_1$ has been computed from the articles with PubMed identifiers 9629849, 8239569 (being referred to by the $P_1, P_2$ provenance identifiers respectively); (ii) the statement $S_2$ has been computed from articles with PubMed identifiers 9629849, 21832222, 22076870 (being referred to by the $P_1, P_3, P_4$ provenance identifiers, respectively[5]). This corresponds to the following indices:*

[5] In reality, the number of source article used for computing these statements in Parkinson's disease knowledge base is much larger, but here we take into account only a few of them to simplify the example.

| *term index* | parkinsonism | DRD | parkinsonisms | mrpi values |
|---|---|---|---|---|
| parkinsonism | 0.0 | 0.545 | 0.0 | 0.0 |
| DRD | 0.545 | 0.0 | 0.0 | 0.0 |
| parkinsonisms | 0.0 | 0.0 | 0.0 | 0.365 |
| mrpi values | 0.0 | 0.0 | 0.365 | 0.0 |

| *statement index* | $S_1$ | $S_2$ |
|---|---|---|
| parkinsonism | 1.0 | 0.0 |
| DRD | 1.0 | 0.0 |
| parkinsonisms | 0.0 | 1.0 |
| mrpi values | 0.0 | 1.0 |

| *provenance index* | $P_1$ | $P_2$ | $P_3$ | $P_4$ |
|---|---|---|---|---|
| $S_1$ | 0.545 | 0.545 | 0.0 | 0.0 |
| $S_2$ | 0.0 | 0.0 | 0.365 | 0.365 |

### Querying

The indices are used to efficiently query for the content of SKIMMR knowledge bases. We currently support atomic queries with one variable, and possibly nested combinations of atomic queries and propositional operators of conjunction (AND), disjunction (OR) and negation (NOT). An atomic query is defined as $? \leftrightarrow T$, where $?$ refers to the query variable and $T$ is a full-text query term.[6] The intended purpose of the atomic query is to retrieve all entities related by any relation to the expressions corresponding to the term $T$. For instance, the $? \leftrightarrow$ parkinsonism query is supposed to retrieve all entities co-occurring-with or similar-to parkinsonism.

[6] One can expand the coverage of their queries using the advanced full-text search features like wildcards or boolean operators for the term look-up. Detailed syntax of the full-text query language we use is provided at http://pythonhosted.org/Whoosh/querylang.html.

Combinations consisting of multiple atomic queries linked by logical operators are evaluated using the following algorithm:

1. Parse the query and generate a corresponding 'query tree' (where each leaf is an atomic query and each node is a logical operator; the levels and branches of this tree reflect the nested structure of the query).

2. Evaluate the atomic queries in the nodes by a look-up in the term index, fetching the term index rows that correspond to the query term in the atomic query.

3. The result of each term look-up is a fuzzy set (*Hájek, 1998*) of terms related to the atomic query term, with membership degrees given by listed weights. One can then naturally combine atomic results by applying fuzzy set operations corresponding to the logical operators in the parsed query tree nodes (where conjunction, disjunction and negation correspond to fuzzy intersection, union and complement, respectively).

4. The result is a fuzzy set of terms $R_T = \{(T_1, w_1^T), (T_2, w_2^T), \ldots, (T_n, w_n^T)\}$, with their membership degrees reflecting their relevance as results of the query.

The term result set $R_T$ can then be used to generate sets of relevant statements ($R_S$) and provenances ($R_P$) using look-ups in the corresponding indices as follows: (a) $R_S = \{(S_1, w_1^S), (S_2, w_2^S), \ldots, (S_m, w_m^S)\}$, where $w_i^S = v_s \sum_{j=1}^{n} w_j^T c_{j,i}$, (b) $R_P = \{(P_1, w_1^P), (P_2, w_2^P), \ldots, (P_q, w_q^P)\}$, where $w_i^P = v_p \sum_{j=1}^{m} w_j^S w_{j,i}$. $v_s, v_p$ are normalisation constants for weights. The weight for a statement $S_i$ in the result set $R_S$ is computed as a normalised a dot product (*i.e.*, sum of the element-wise products) of the vectors given by: (a) the membership degrees in the term result set $R_T$, and (b) the column in the statement index that corresponds to $S_i$. Similarly, the weight for a provenance $P_i$ in the result set $R_P$ is a normalised dot product of the vectors given by the $S_T$ membership degrees and the column in the provenance index corresponding to $P_i$.

The fuzzy membership degrees in the term, statement and provenance result sets can be used for ranking and visualisation, prioritising the most important results when presenting them to the user. The following example outlines how a specific query is evaluated.

**Example 5** *Let us assume we want to query the full SKIMMR knowledge base about Parkinson's Disease for the following:*

? ↔ parkinsonism AND (? ↔ mrpi OR ? ↔ magnetic resonance parkinsonism index)

*This aims to find all statements (and corresponding documents) that are related to* parkinsonism *and either* magnetic resonance parkinsonism index *or its* mrpi *abbreviation. First of all, the full-text index is queried, retrieving two different terms conforming to the first atomic part of the query due to its stemming features:* parkinsonism *and* parkinsonisms. *The other two atomic parts of the initial query are resolved as is. After the look-up in the term index, four fuzzy sets are retrieved: 1.* $T_{\text{parkinsonism}}$ *(3,714 results), 2.* $T_{\text{parkinsonisms}}$ *(151 results), 3.* $T_{\text{mrpi}}$ *(39 results). 4.* $T_{\text{magnetic resonance parkinsonism index}}$ *(29 results). The set of terms conforming to the query is then computed as*

$(T_{\text{parkinsonism}} \cup T_{\text{parkinsonisms}}) \cap (T_{\text{mrpi}} \cup T_{\text{magnetic resonance parkinsonism index}}).$

*When using maximum and minimum as t-conorm and t-norm for computing the fuzzy union and intersection (*Hájek, 1998*), respectively, the resulting set has 29 elements with non-zero membership degrees. The top five of them are*

*(1)* cup, *(2)* mrpi, *(3)* magnetic resonance parkinsonism index, *(4)* clinically unclassifiable parkinsonism, *(5)* clinical evolution

*with membership degrees* 1.0, 1.0, 0.704, 0.39, 0.34, *respectively. According to the statement index, there are* 138 *statements corresponding to the top five term results of the initial query, composed of* 136 *co-occurrences and* 2 *similarities. The top five co-occurrence statements and the two similarity statements are:*

| Type | Entity$_1$ | Entity$_2$ | Membership degree |
|------|-----------|-----------|-------------------|
| *cooc* | `mrpi` | `cup` | 1.0 |
| *cooc* | `mrpi` | `magnetic resonance parkinsonism index` | 0.852 |
| *cooc* | `cup` | `magnetic resonance parkinsonism index` | 0.852 |
| *cooc* | `mrpi` | `clinically unclassifiable parkinsonism` | 0.695 |
| *cooc* | `cup` | `clinically unclassifiable parkinsonism` | 0.695 |
| *sim* | `psp patients` | `magnetic resonance parkinsonism index` | 0.167 |
| *sim* | `parkinsonism` | `clinical evolution` | 0.069 |

*where the membership degrees are computed from the combination of the term weights as described before the example, using an arithmetic mean for the aggregation. Finally, a look-up in the source index for publications corresponding to the top seven result statements retrieves 8 relevant PubMed identifiers (PMID). The top five of them correspond to the following list of articles:*

| PMID | Title | Authors | Weight |
|------|-------|---------|--------|
| 21832222 | The diagnosis of neurodegenerative disorders based on clinical and pathological findings using an MRI approach | Watanabe H et al. | 1.0 |
| 21287599 | MRI measurements predict PSP in unclassifiable parkinsonisms: a cohort study | Morelli M et al. | 0.132 |
| 22277395 | Accuracy of magnetic resonance parkinsonism index for differentiation of progressive supranuclear palsy from probable or possible Parkinson disease | Morelli M et al. | 0.005 |
| 15207208 | Utility of dopamine transporter imaging (123-I Ioflupane SPECT) in the assessment of movement disorders | Garcia Vicente AM et al. | 0.003 |
| 8397761 | Alzheimer's disease and idiopathic Parkinson's disease coexistence | Rajput AH et al. | 0.002 |

*where the weights have been computed by summing up the statement set membership degrees multiplied by the source index weights and then normalising the values by their maximum.*

## Evaluation methodology

In addition to proposing specific methods for creating knowledge bases that support skim reading, we have also come up with a specific methodology for evaluating the generated knowledge bases. An ideal method for evaluating the proposed approach, implemented as a SKIMMR tool, would be to record and analyse user feedback and behaviour via SKIMMR instances used by large numbers of human experts. We do have such means for evaluating SKIMMR implemented in the user interface.[7] However, we have not yet managed to collect sufficiently large sample of user data due to the early stage of the prototype deployment. Therefore we implemented an indirect methodology for automated quantitative evaluation of SKIMMR instances using publicly available manually

[7] See the SMA SKIMMR instance at http://www.skimmr.org:8008/data/html/trial.tmp for details.

curated data. The methodology is primarily based on simulation of various types of human behaviour when browsing the entity networks generated by SKIMMR. We formally define certain properties of the simulations and measure their values in order to determine the utility of the entity networks for the purposes of skim reading. Details are given in the following sections.

### Overview of the evaluation methods

The proposed methods intend to simulate human behaviour when using the data generated by SKIMMR. We apply the same simulations also to baseline data that can serve for the same or similar purpose as SKIMMR (i.e., discovery of new knowledge by navigating entity networks). Each simulation is associated with specific measures of performance, which can be used to compare the utility of SKIMMR with respect to the baseline.

The primary evaluation method is based on random walks (*Lovász, 1993*) in an undirected entity graph corresponding to the SKIMMR knowledge base. For the baseline, we use a network of MeSH terms assigned by human curators to the PubMed abstracts that have been used to create the SKIMMR knowledge base.[8] This represents a very similar type of content, i.e., entities associated with PubMed articles. It is also based on expert manual annotations and thus supposed to be a reliable gold standard (or at least a decent approximation thereof due to some level of transformation necessary to generate the entity network from the annotations).

**Example 6** *Returning to the knowledge base statement from* Example 2 *in 'Corpus-wide co-occurrence':* (`parkinsonism,cooc,DRD,0.545`)*. In the SKIMMR entity graph, this corresponds to two nodes* (`parkinsonism,DRD`) *and one edge between them with weight 0.545. We do not distinguish between the types of the edges (i.e., co-occurrence or similarity), since it is not of significant importance for the SKIMMR users according to our experience so far (they are more interested in navigating the connections between nodes regardless of the connection type).*

*A baseline entity graph is generated from the PubMed annotations with MeSH terms. For all entities* $X, Y$ *associated with an abstract* $A$*, we construct an edge connecting the nodes* $X$ *and* $Y$ *in the entity graph. The weight is implicitly assumed to be 1.0 for all such edges. To explain this using concrete data, let us consider the two PubMed IDs from* Example 1*,* 9629849 *and* 8239569*. Selected terms from the corresponding MeSH annotations are* {`Parkinson Disease/radionuclide imaging,Male,Child`}*,* {`Parkinson Disease/radionuclide imaging,Dystonia/drug therapy`}*, respectively. The graph induced by these annotations is depicted in* Fig. 1*.*

The secondary evaluation method uses an index of related articles derived from the entities in the SKIMMR knowledge bases. For the baseline, we use either an index of related articles produced by a specific service of PubMed (*Lin & Wilbur, 2007*), or the evaluation data from the document categorisation task of the TREC'04 genomics track (*Cohen & Hersh, 2006*) where applicable. We use the TREC data since they were used also for evaluation of the actual algorithm used by PubMed to compute related articles.

---

[8] MeSH (Medical Subject Headings) is a comprehensive, manually curated and regularly updated controlled vocabulary and taxonomy of biomedical terms. It is frequently used as a standard for annotation of biomedical resources, such as PubMed abstracts. See http://www.ncbi.nlm.nih.gov/mesh for details.

**Peer**J

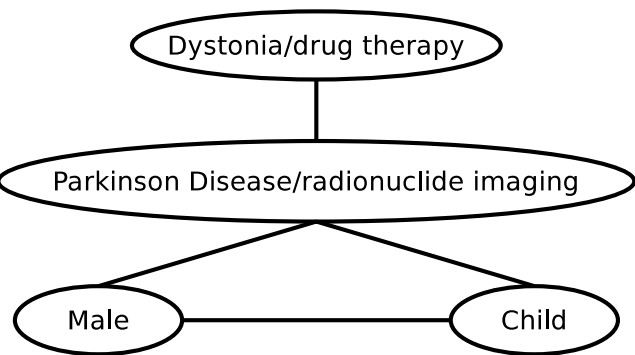

**Figure 1** **Example of an entity graph derived from PubMed.**

To generate the index of related articles from the SKIMMR data, we first use the knowledge base indices (see 'Extracting basic co-occurrence statements from texts') to generate a mapping $E_P : E \rightarrow 2^P$ from entities from a set $E$ to a set of corresponding provenance identifiers (subsets of a set $P$). In the next step, we traverse the entity graph $G_E$ derived from the statements in the SKIMMR knowledge base and build an index of related articles according to the following algorithm:

1. Initialise a map $M_P$ between all possible $(P_i, P_j)$ provenance identifier pairs and the weight of an edge between them so that all values are zero.
2. For all pairs of entities $E_1, E_n$ (i.e., nodes in $G_E$), do:
   - If there is a path $\mathcal{P}$ of edges $\{(E_1, E_2), (E_2, E_3), \ldots, (E_{n-1}, E_n)\}$ in $G_E$:
     - compute an aggregate weight of the path as $w_\mathcal{P} = w_{E_1, E_2} \cdot w_{E_2, E_3} \cdot \ldots \cdot w_{E_{n-1}, E_n}$ (as a multiplication of all weights along the path $\mathcal{P}$);
     - set the values $M_P(P_i, P_j)$ of the map $M_P$ to $max(M_P(P_i, P_j), w_\mathcal{P})$ for every $P_i, P_j$ such that $P_i \in E_P(E_1), P_j \in E_P(E_n)$ (i.e., publications corresponding to the source and target entities of the path).
3. Interpret the $M_P$ map as an adjacency matrix and construct a corresponding weighted undirected graph $G_P$.
4. For every node $P$ in $G_P$, iteratively construct the index of related articles by associating the key $P$ with a list $L$ of all neighbours of $P$ in $G_P$ sorted by the weights of the corresponding edges.

Note that in practice, we restrict the maximum length of the paths to three and also remove edges in $G_P$ with weight below 0.1. This is to prevent a combinatorial explosion of the provenance graph when the entity graph is very densely connected.

The baseline index of related publications according to the PubMed service is simply a mapping of one PubMed ID to an ordered list of the related PubMed IDs. The index based on the TREC data is generated from the article categories in the data set. For a PubMed ID $X$, the list of related IDs are all IDs belonging to the same category as $X$, ordered so that the definitely relevant articles occur before the possibly relevant ones.[9]

[9] The articles in the TREC data set are annotated by membership in a number of specific categories. The membership is gradual, with three possible values–definitely relevant, possibly relevant and not relevant.

### Motivation of the evaluation methods

The random walks are meant to simulate user's behaviour when browsing the SKIMMR data, starting with an arbitrary entry point, traversing a number of edges linking the entities and ending up in a target point. Totally random walk corresponds to when a user browses randomly and tries to learn something interesting along the way. Other types of user behaviour can be simulated by introducing specific heuristics for selection of the next entity on the walk (see below for details). To determine how useful a random walk can be, we measure properties like the amount of information along the walk and in its neighbourhood, or semantic similarity between the source and target entities (i.e., how semantically coherent the walk is).

The index of related articles has been chosen as a secondary means for evaluating SKIMMR. Producing links between publications is not the main purpose of our current work, however, it is closely related to the notion of skim reading. Furthermore, there are directly applicable gold standards we can use for automated evaluation of the lists of related articles generated by SKIMMR, which can provide additional perspective on the utility of the underlying data even if we do not momentarily expose the publication networks to users.

### Running and measuring the random walks

To evaluate the properties of random walks in a comprehensive manner, we ran them in batches with different settings of various parameters. These are namely: (1) *heuristics* for selecting the next entity (one of the four defined below); (2) *length* of the walk (2, 5, 10 or 50 edges); (3) *radius* of the walk's envelope, i.e., the maximum distance between the nodes of the path and entities that are considered its neighbourhood (0, 1, 2); (4) number of *repetitions* (100-times for each combination of the parameter (1–3) settings).

Before we continue, we have to introduce few notions that are essential for the definition of the random walk heuristics and measurements. The first of them is a set of top-level (abstract) clusters associated with an entity in a graph (either from SKIMMR or from PubMed) according to the MeSH taxonomy. This is defined as a function $C_A : E \to M$, where $E, M$ are the sets of entities and MeSH cluster identifiers, respectively. The second notion is a set of specific entity cluster identifiers $C_S$, defined on the same domain and range as $C_A$, *i.e.*, $C_S : E \to M$.

The MeSH cluster identifiers are derived from the tree path codes associated with each term represented in MeSH. The tree path codes have the form $L_1.L_2. \ldots .L_{n-1}.L_n$ where $L_i$ are sub-codes of increasing specificity (i.e., $L_1$ is the most general and $L_n$ most specific). For the abstract cluster identifiers, we take only the top-level tree path codes into account as the values of $C_A$, while for $C_S$ we consider the complete codes. Note that for the automatically extracted entity names in SKIMMR, there are often no direct matches in the MeSH taxonomy that could be used to assign the cluster identifiers. In these situations, we try to find a match for the terms and their sub-terms using a lemmatised full-text index implemented on the top of MeSH. This helps to increase the coverage two- to three-fold on our experimental data sets.

For some required measures, we will need to consider the number and size of specific clusters associated with the nodes in random walks and their envelopes. Let us assume a set of entities $Z \subseteq E$. The number of clusters associated with the entities from $Z$, $cn(Z)$, is then defined as $cn(Z) = |\bigcup_{X \in Z} C(X)|$ where $C$ is one of $C_A, C_S$ (depending on which type of clusters are we interested in). The size of a cluster $C_i \in C(X)$, $cs(C_i)$, is defined as an absolute frequency of the mentions of $C_i$ among the clusters associated with the entities in $Z$. More formally, $cs(C_i) = |\{X | X \in Z \wedge C_i \in C(X)\}|$. Finally, we need a MeSH-based semantic similarity of entities $sim_M(X, Y)$, which is defined in detail in the formula (8) in 'Similarities'.

**Example 7** *To illustrate the MeSH-based cluster annotations and similarities, let us consider two entities,* `supranuclear palsy, progressive, 3` *and* `secondary parkinson disease`. *The terms correspond to the MeSH tree code sets* {C10.228.662.700, ..., C23.888.592.636.447.690, ..., C11.590.472.500, ...} *and* {C10.228.662.600.700}, *respectively, which are also the sets of specific clusters associated with the terms. The top-level clusters are* {C10, C11, C23} *and* {C10}, *respectively. The least common subsumer of the two terms is* C10.228.662 *of depth 3 (the only possibility with anything in common is* C10.228.662.700 *and* C10.228.662.600.700). *The depths of the related cluster annotations are 4 and 5, therefore the semantic similarity is* $\frac{2 \cdot 3}{4+5} = \frac{2}{3}$.

We define four heuristics used in our random walk implementations. All the heuristics select the next node to visit in the entity graph according to the following algorithm:

1. Generate the list $L$ of neighbours of the current node.
2. Sort $L$ according to certain criteria (heuristic-dependent).
3. Initialise a threshold $e$ to $e_i$, a pre-defined number in the $(0, 1)$ range (we use 0.9 in our experiments).
4. For each node $u$ in the sorted list $L$, do:
   - Generate a random number $r$ from the $[0, 1]$ range.
   - If $r \leq e$:
     – return $u$ as the next node to visit.
   - Else:
     – set $e$ to $e \cdot e_i$ and continue with the next node in $L$.
5. If nothing has been selected by now, return a random node from $L$.

All the heuristics effectively select the nodes closer to the head of the sorted neighbour list more likely than the ones closer to the tail. The random factor is introduced to emulate the human way of selecting next nodes to follow, which is often rather fuzzy according to our observations of sample SKIMMR users.

The distinguishing factor of the heuristics are the criteria for sorting the neighbour list. We employed the following four criteria in our experiments: (1) giving preference to the nodes that have not been visited before ($H = 1$); (2) giving preference to the nodes connected by edges with higher weight ($H = 2$); (3) giving preference to the nodes that are

more similar, using the $sim_M$ function introduced before ($H = 3$); (4) giving preference to the nodes that are less similar ($H = 4$). The first heuristic simulates a user that browses the graph more or less randomly, but prefers to visit previously unknown nodes. The second heuristic models a user that prefers to follow a certain topic (i.e., focused browsing). The third heuristic represents a user that wants to learn as much as possible about many diverse topics. Finally, the fourth heuristic emulates a user that prefers to follow more plausible paths (approximated by the weight of the statements computed by SKIMMR).

Each random walk and its envelope (i.e., the neighbourhood of the corresponding paths in the entity graphs) can be associated with various information-theoretic measures, graph structure coefficients, levels of correspondence with external knowledge bases, etc. Out of the multitude of possibilities, we selected several specific scores we believe to soundly estimate the value of the underlying data for users in the context of skim reading.

Firstly, we measure **semantic coherence** of the walks. This is done using the MeSH-based semantic similarity between the nodes of the walk. In particular, we measure: (A) coherence between the source $S$ and target $T$ nodes as $sim_M(S, T)$; (B) product coherence between all the nodes $U_1, U_2, \ldots, U_n$ of the walk as $\Pi_{i \in \{1, \ldots, n-1\}} sim_M(U_i, U_{i+1})$; (C) average coherence between all the nodes $U_1, U_2, \ldots, U_n$ of the walk as $\frac{1}{n} \sum_{i \in \{1, \ldots, n-1\}} sim_M(U_i, U_{i+1})$. This family of measures helps us to assess how convergent (or divergent) are the walks in terms of focus on a specific topic.

The second measure we used is the **information content** of the nodes on and along the walks. For this, we use the entropy of the association of the nodes with clusters defined either (a) by the MeSH annotations or (b) by the structure of the envelope. By definition, the higher the entropy of a variable, the more information the variable contains (*Shannon, 1948*). In our context, a high entropy value associated with a random walk means that there is a lot of information available for the user to possibly learn when browsing the graph. The specific entropy measures we use relate to the following sets of nodes: (D) abstract MeSH clusters, path only; (E) specific MeSH clusters, path only; (F) abstract MeSH clusters, path and envelope; (G) specific MeSH clusters, path and envelope; (H) clusters defined by biconnected components (*Hopcroft & Tarjan, 1973*) in the envelope.[10] The entropies of the sets (D–G) are defined by formulae (9) and (10) in 'Entropies'.

The last family of random walk evaluation measures is based on the **graph structure** of the envelopes: (I) envelope size (in nodes); (J) envelope size in biconnected components; (K) average component size (in nodes); (L) envelope's clustering coefficient. The first three measures are rather simple statistics of the envelope graph. The clustering coefficient is widely used as a convenient scalar representation of the structural complexity of a graph, especially in the field of social network analysis (*Carrington, Scott & Wasserman, 2005*). In our context, we can see it as an indication of how likely it is that the connections in the entity graph represent non-trivial relationships.

To facilitate the interpretation of the results, we computed also the following auxiliary measures: (M) number of abstract clusters along the path; (N) average size of the abstract clusters along the path; (O) number of abstract clusters in the envelope; (P) average size of the abstract clusters in the envelope; (Q) number of specific clusters along the path;

[10] Biconnected components can be understood as sets of nodes in a graph that are locally strongly connected and therefore provide us with a simple approximation of clustering in the entity graphs based purely on their structural properties.

(R) average size of the specific clusters along the path; (S) number of specific clusters in the envelope; (T) average size of the specific clusters in the envelope. Note that all the auxiliary measures use the MeSH cluster size and number notions, i.e., $cs(\ldots)$ and $cn(\ldots)$ as defined earlier.

### Comparing the indices of related articles

The indices of related articles have quite a simple structure. We can also use the baseline indices as gold standard, and therefore evaluate the publication networks implied by the SKIMMR data using classical measures of precision and recall (*Manning, Raghavan & Schütze, 2008*). Moreover, we can also compute correlation between the ranking of the items in the lists of related articles which provides an indication of how well SKIMMR preserves the ranking imposed by the gold standard.

For the correlation, we use the standard Pearson's formula (*Dowdy, Weardon & Chilko, 2005*), taking into account only the ranking of articles occurring in both lists. The measures of precision and recall are defined using overlaps of the sets of related articles in the SKIMMR and gold standard indices. The detailed definitions of the specific notions of precision and recall we use are given in formulae (11) and (12) in 'Precision and recall'. The gold standard is selected depending on the experimental data set, as explained in the next section. In order to cancel out the influence of different average lengths of lists of related publications between the SKIMMR and gold standard indices, one can take into account only a limited number of the most relevant (i.e., top) elements in each list.

## RESULTS

We have implemented the techniques described in the previous section as a set of software modules and provided them with a search and browse front-end. This forms a prototype implementation of SKIMMR, available as an open source software package through the GitHub repository (see 'Software packages' for details). We here describe the architecture of the SKIMMR software ('Architecture') and give examples on the typical use of SKIMMR in the domains of Spinal Muscular Atrophy and Parkinson's Disease ('Using SKIMMR'). 'Evaluation' presents an evaluation of the proposed approach to machine-aided skim reading using SKIMMR running on three domain-specific sets of biomedical articles.

### Architecture

The SKIMMR architecture and data flow is depicted in Fig. 2. First of all, SKIMMR needs a list of PubMed identifiers (unique numeric references to articles indexed on PubMed) specified by the user of system administrator. Then it automatically downloads the abstracts of the corresponding articles and stores the texts locally. Alternatively, one can export results of a manual PubMed search as an XML file (using the '*send to file*' feature) and then use a SKIMMR script to generate text from that file. From the texts, a domain-specific SKIMMR knowledge base is created using the methods described in 'Extracting basic co-occurrence statements from texts' and 'Computing a knowledge base from the extracted statements'. The computed statements and their article provenance are then indexed as described in 'Indexing and querying the knowledge base'. This allows

**Peer**J

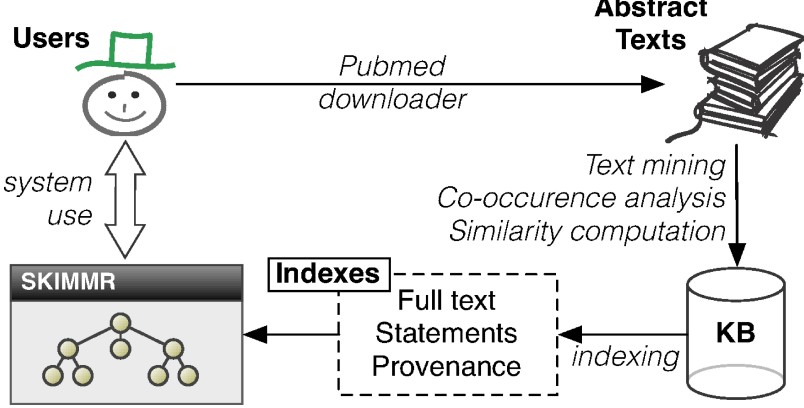

**Figure 2** **Architecture of the SKIMMR system.**

users to search and browse the high-level graph summaries of the interconnected pieces of knowledge in the input articles. The degrees in the result sets (explained in detail in 'Indexing and querying the knowledge base') are used in the user interface to prioritise the more important nodes in the graphs by making their font and size proportional to the sum of the degrees of links (i.e., the number of statements) associated with them. Also, only a selected amount of the top scoring entities and links between them is displayed at a time.

## Using SKIMMR

The general process of user interaction with SKIMMR can be schematically described as follows:

1. Search for an initial term of interest in a simple query text box.
2. A graph corresponding to the results of the search is displayed. The user has two options then:
   (a) Follow a link to another node in the graph, essentially browsing the underlying knowledge base along the chosen path by displaying the search results corresponding to the selected node and thus going back to step 1 above.
   (b) Display most relevant publications that have been used for computing the content of the result graph, going to step 3 below.
3. Access and study the displayed publications in detail using a re-direct to PubMed.

The following two sections illustrate the process using examples from two live instances of SKIMMR deployed on articles about Spinal Muscular Atrophy and Parkinson's Disease.[11] The last section of this part of the article gives a brief overview of the open source software packages of SKIMMR available for developers and users interested in deploying SKIMMR on their own data.

[11] The live instances are running at http://www.skimmr.org:8008 and http://www.skimmr.org:8090, respectively, as of June 2014. Canned back-up versions of them are available at http://www.skimmr.org/resources/skimmr/sma.tgz and http://www.skimmr.org/resources/skimmr/pd.tgz (SMA and Parkinson's Disease, respectively). If the SKIMMR dependencies are met (see https://github.com/vitnov/SKIMMR), the canned instances can be used locally on any machine with Python installed (versions higher than 2.4 and lower than 3.0 are supported, while 2.6.\* and 2.7.\* probably work best). After downloading the archives, unpack them and switch to the resulting folder. Run the re-indexing script, following Section 3.6 in the README provided in the same folder. To execute the SKIMMR front-end locally, run the server as described in Section 3.7 of the README.

**Peer**J

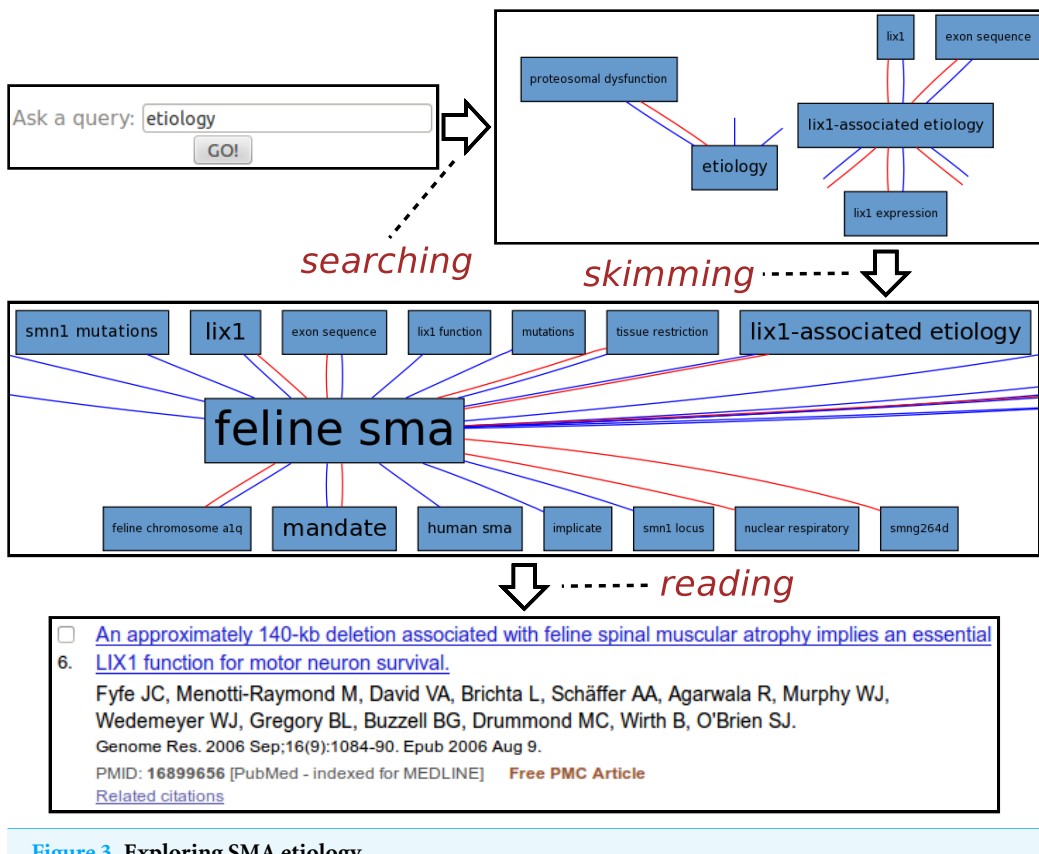

**Figure 3** Exploring SMA etiology.

### Spinal muscular atrophy

Fig. 3 illustrates a typical session with the Spinal Muscular Atrophy[12] instance of SKIMMR. The SMA instance was deployed on a corpus of 1,221 abstracts of articles compiled by SMA experts from the SMA foundation.[13]

The usage example is based on an actual session with Maryann Martone, a neuroscience professor from UCSD and a representative of the SMA Foundation who helped us to assess the potential of the SKIMMR prototype. Following the general template from the beginning of the section, the SMA session can be divided into three distinct phases:

1. **Searching:** The user was interested in the SMA etiology (studies on underlying causes of a disease). The key word `etiology` was thus entered into the search box.

2. **Skimming:** The resulting graph suggests relations between etiology of SMA, various gene mutations, and the Lix1 gene. Lix1 is responsible for protein expression in limbs which seems relevant to the SMA manifestation, therefore the `Lix1 — associated etiology` path was followed in the graph, moving on to a slightly different area in the underlying knowledge base extracted from the SMA abstracts. When browsing the graph along that path, one can quickly notice recurring associations with `feline SMA`. According to the neuroscience expert we consulted, the cat models of the SMA disease appear to be quite a specific and interesting fringe area of SMA

[12] A genetic neurological disease caused by mutation of SMN1 gene that leads to death of motor neurons and consequent progressive muscle atrophy. It is the most common genetic cause of infant death and there is no cure as of now. See http://en.wikipedia.org/wiki/Spinal_muscular_atrophy for details.

[13] See http://www.smafoundation.org/.

research. Related articles may be relevant and enlightening even for experienced researchers in the field.

3. **Reading:** The reading mode of SKIMMR employs an in-line redirect to a specific PubMed result page. This way one can use the full set of PubMed features for exploring and reading the articles that are mostly relevant to the focused area of the graph the user skimmed until now. The sixth publication in the result was most relevant for our sample user, as it provided more details on the relationships between a particular gene mutation in a feline SMA model and the Lix1 function for motor neuron survival. This knowledge, albeit not directly related to SMA etiology in humans, was deemed as enlightening by the domain expert in the context of the general search for the culprits of the disease.

The whole session with the neuroscience expert lasted about two minutes and clearly demonstrated the potential for serendipitous knowledge discovery with our tool.

### Parkinson's disease

Another example of the usage of SKIMMR is based on a corpus of 4,727 abstracts concerned with the clinical studies of Parkinson's Disease (PD). A sample session with the PD instance of SKIMMR is illustrated in Fig. 4. Following the general template from the beginning of the section, the PD session can be divided into three distinct phases again:

1. **Searching:** The session starts with typing `parkinson's` into the search box, aiming to explore the articles from a very general entry point.

2. **Skimming:** After a short interaction with SKIMMR, consisting of few skimming steps (i.e., following a certain path in the underlying graphs of entities extracted from the PD articles), an interesting area in the graph has been found. The area is concerned with `Magnetic Resonance Parkinsons Index` (MRPI). This is a numeric score calculated by multiplying two structural ratios: one for the area of the pons relative to that of the midbrain and the other for the width of the Middle Cerebellar Peduncle relative to the width of the Superior Cerebellar Peduncle. The score is used to diagnose PD based on neuroimaging data (*Morelli et al., 2011*).

3. **Reading:** When displaying the articles that were used to compute the subgraph surrounding MRPI, the user reverted to actual reading of the literature concerning MRPI and related MRI measures used to diagnose Parskinson's Disease as well a range of related neurodegenerative disorders.

This example illustrates once again how SKIMMR provides an easy way of navigating through the conceptual space of a subject that is accessible even to novices, reaching interesting and well-specified components areas of the space very quickly.

### Software packages

In addition to the two live instances described in the previous sections, SKIMMR is available for local installation and custom deployment either on biomedical article abstracts from PubMed, or on general English texts. Moreover, one can expose SKIMMR

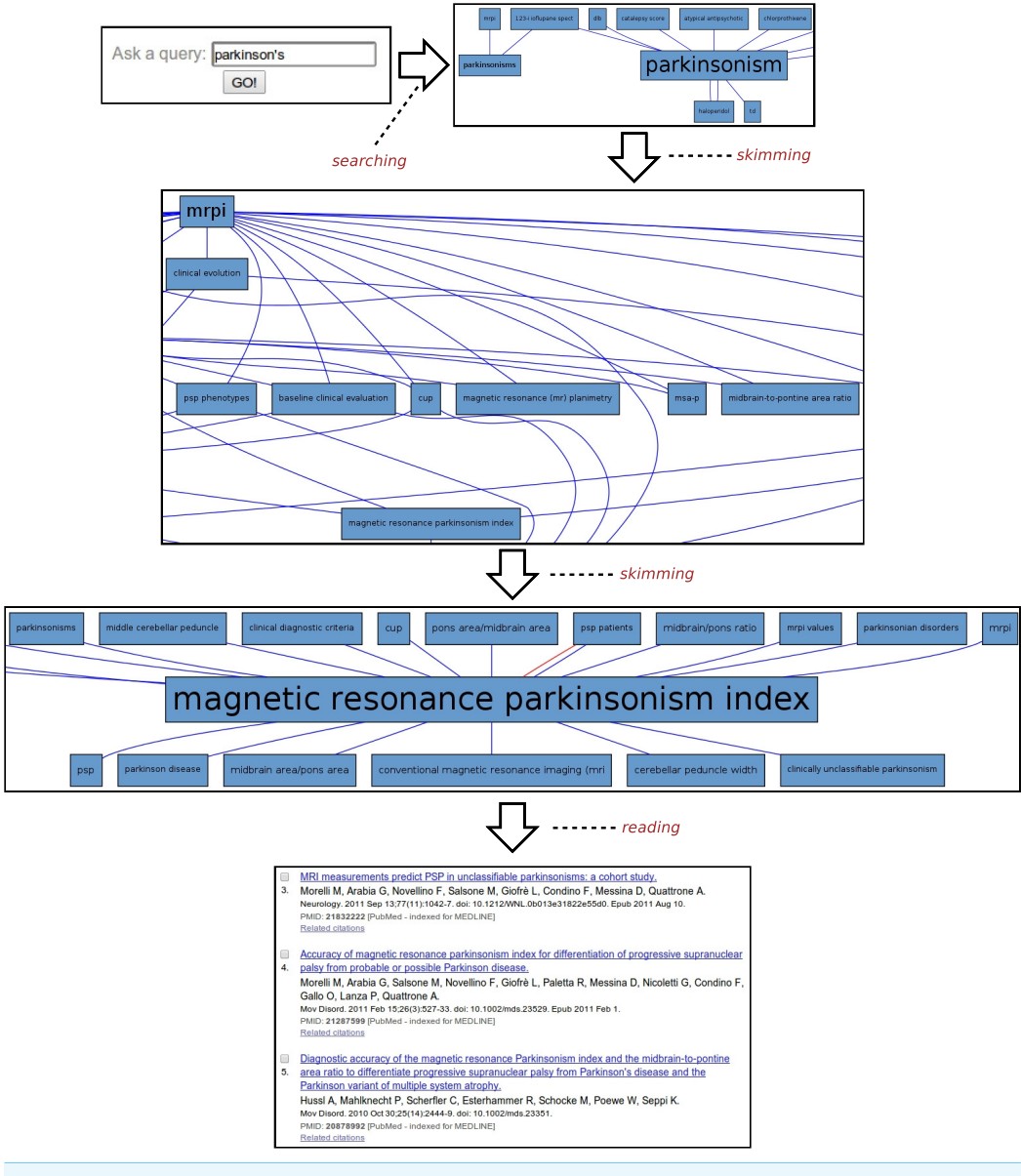

**Figure 4  Exploring Parkinson's disease.**

via a simple HTTP web service once the back-end has compiled a knowledge base from selected textual input. The latter is particularly useful for the development of other applications on the top of the content generated by SKIMMR. Open source development snapshots (written in the Python programming language) of SKIMMR modules are available via our GitHub repository[14] with accompanying documentation.

## Evaluation

In the following we report on experiments we used for evaluating SKIMMR using the method explained in 'Evaluation methodology'. The results of our experiments empirically demonstrate that the SKIMMR networks allow for more focused browsing

[14] See https://github.com/vitnov/SKIMMR.

of the publication content than is possible with tools like PubMed. SKIMMR also has the potential for offering more information of higher complexity during the browsing process. The following sections provide details on the data sets used in the experiments and the results of the evaluation.

### Evaluation data

We have evaluated SKIMMR using three corpora of domain-specific biomedical articles. The first one was SMA: a representative corpus of 1,221 PubMed abstracts dealing with Spinal Muscular Atrophy (SMA), compiled by experts from SMA Foundation. The second corpus was PD: a set of 4,727 abstracts that came as results (in February 2013) of a search for clinical studies on Parkinson's Disease on PubMed. The last corpus was TREC: a random sample[15] of 2,247 PubMed abstracts from the evaluation corpus of the TREC'04 genomics track (document categorisation task).

For running the experiment with random walks, we generated two graphs for each of the corpora (using the methods described in Example 6): (1) network of SKIMMR entities; (2) network of MeSH terms based on the PubMed annotations of the articles that were used as sources for the particular SKIMMR instance.

As outlined before in the methods section, we also used some auxiliary data structures for the evaluation. The first auxiliary resource was the MeSH thesaurus (version from 2013). From the data available on the National Library of Medicine web site, we generated a mapping from all MeSH terms and their synonyms to the corresponding tree codes indicating their position in the MeSH hierarchy. We also implemented a lemmatised full-text index on the MeSH mapping keys to increase the coverage of the tree annotations when the extracted entity names do not exactly correspond to the MeSH terms.

The second type of auxiliary resource (a gold standard) were indices of related articles based on the corresponding PubMed service. For the other type of gold standard, we used the TREC'04 category associations from the genomics track data. This is essentially a mapping between PubMed IDs, category identifiers and a degree of membership of the specific IDs in the category (definitely relevant, possibly relevant, not relevant). From that mapping, we generated the index of related articles as a gold standard for the secondary evaluation method (the details of the process are described in the previous section).

Note that for the TREC corpus, the index of related articles based on the TREC data is applicable as a gold standard for the secondary evaluation. However, for the other two data sets (SMA and PD), we used the gold standard based on the PubMed service for fetching related articles. This is due to almost zero overlap between the TREC PubMed IDs and the SMA, PD corpora, respectively.

### Data statistics

*Corpus and knowledge base statistics.* Basic statistics of the particular text corpora are given in Table 1, with column explanations as follows: (1) |*SRC*| is the number of the source documents; (2) |*TOK*| is the number of tokens (words) in the source documents; (3) |*BC*| is the number of base co-occurrence statements extracted from the sources (see 'Extracting basic co-occurrence statements from texts' for details); (4) |*LEX*| is the vocabulary size

---

[15] We processed only a subset of the experimental data available from TREC so that the experimental knowledge bases are of a size within similar range of hundreds of thousands of statements.

**Table 1  Basic statistics of the SKIMMR instances.**

| Data set ID | $|SRC|$ | $|TOK|$ | $|BC|$ | $|LEX|$ | $|KB_{cooc}|$ | $|KB_{sim}|$ |
|---|---|---|---|---|---|---|
| SMA | 1,221 | 223,257 | 333,124 | 15,288 | 308,626 | 23,167 |
| PD | 4,727 | 943,444 | 1,096,037 | 43,410 | 965,753 | 57,876 |
| TREC | 2,247 | 439,202 | 757,762 | 39,431 | 745,201 | 65,510 |

**Table 2  Derived statistics of the SKIMMR instances.**

| Data set ID | $T/S$ | $B/S$ | $L/T$ | $SM/KB$ | $KB/S$ | $KB/L$ |
|---|---|---|---|---|---|---|
| SMA | 182.848 | 272.829 | 0.068 | 0.07 | 271.739 | 21.703 |
| PD | 199.586 | 231.867 | 0.046 | 0.057 | 216.549 | 23.58 |
| TREC | 195.462 | 337.233 | 0.09 | 0.081 | 360.797 | 20.56 |

(i.e., the number of unique entities occurring in the basic co-occurrence statements); (5) $|KB_{cooc}|$ is the number of aggregate co-occurrence statements in the corresponding SKIMMR knowledge base (see 'Corpus-wide co-occurrence'); (6) $|KB_{sim}|$ is the number of similarity statements in the corresponding SKIMMR knowledge base (see 'Similarity').

Derived statistics on the SKIMMR instances are provided in Table 2, with column explanations as follows: (1) $T/S$ is an average number of tokens per a source document; (2) $B/S$ is an average number of basic co-occurrence statements per a source document; (3) $L/T$ is a ratio of the size of the lexicon with respect to the overall number of tokens in the input data; (4) $SM/KB$ is a ratio of the similarity statements to the all statements in the knowledge base; (5) $KB/S$ is an average number of statements in the knowledge base per a source document; (6) $KB/L$ is an average number of statements in the knowledge base per a term in the lexicon. The values in the columns are computed from the basic statistics as follows:

$$T/S = \frac{|TOK|}{|SRC|}, \qquad B/S = \frac{|BC|}{|SRC|}, \qquad L/T = \frac{|LEX|}{|TOK|}, \qquad SM/KB = \frac{|KB_{sim}|}{|KB_{sim}| + |KB_{cooc}|},$$

$$KB/S = \frac{|KB_{sim}| + |KB_{cooc}|}{|SRC|}, \qquad KB/L = \frac{|KB_{sim}| + |KB_{cooc}|}{|LEX|}.$$

The statistics of the data sets are relatively homogeneous. The TREC data contains more base co-occurrence statements per article, and has an increased ratio of (unique) lexicon terms per absolute number of (non-unique) tokens in the documents. TREC knowledge base also contains more statements per article than the other two, but the ratios of number of statements in it per lexicon term are more or less balanced. We believe that the statistics do not imply the need to treat each of the data sets differently when interpreting the results reported in the next section.

*Graph statistics.* The statistics of the graph data that are utilised in the random walks experiment are given in Tables 3 and 4 for PubMed and SKIMMR, respectively. The specific

**Table 3** Statistics of the PubMed graphs for random walks.

| Data set ID | $|V|$ | $|E|$ | $\frac{|E|}{|V|}$ | $D$ | $d$ | $l_G$ | $|C|$ |
|---|---|---|---|---|---|---|---|
| SMA | 5,364 | 78,608 | 14.655 | $5.465 \cdot 10^{-3}$ | 5.971 | 3.029 | 2 |
| PD | 8,622 | 133,188 | 15.447 | $3.584 \cdot 10^{-3}$ | 6 | 2.899 | 2 |
| TREC | 10,734 | 161,838 | 15.077 | $2.809 \cdot 10^{-3}$ | 7.984 | 3.146 | 3 |

**Table 4** Statistics of the SKIMMR graphs for random walks.

| Data set ID | $|V|$ | $|E|$ | $\frac{|E|}{|V|}$ | $D$ | $d$ | $l_G$ | $|C|$ |
|---|---|---|---|---|---|---|---|
| SMA | 15,287 | 305,077 | 19.957 | $2.611 \cdot 10^{-3}$ | 5 | 2.642 | 1 |
| PD | 43,411 | 952,296 | 21.937 | $1.011 \cdot 10^{-3}$ | 5 | 2.271 | 2 |
| TREC | 37,184 | 745,078 | 20.038 | $1.078 \cdot 10^{-3}$ | 5.991 | 2.999 | 12 |

[16] Note that the number of edges is lower in the SKIMMR graphs than in the corresponding SKIMMR knowledge bases due to the fact that we do not distinguish between the different relationships. Therefore, if two nodes are connected by more than one statements, there is still only one edge for those nodes in the graph.

statistics provided on the graphs are: (1) number of nodes ($|V|$); (2) number of edges[16] ($|E|$); (3) average number of edges per a node ($\frac{|E|}{|V|}$); (4) density ($D = \frac{2 \cdot |E|}{|V|(|V|-1)}$, i.e., a ratio of the actual bidirectional connections between nodes relative to the maximum possible number of connections); (5) diameter ($d$, computed as an arithmetic mean of the longest possible paths in the connected components of the graph, weighted by the size of the components in nodes); (6) average shortest path length ($l_G$, computed similarly to $d$ as an average weighted mean of the value for each connected component); (7) number of connected components ($|C|$).

The statistics demonstrate that the SKIMMR graphs are larger and have higher absolute number of connections per a node, but are less dense than the PubMed graphs. All the graphs exhibit the "small-world" property (*Watts & Strogatz, 1998*), since the graphs have small diameters and there are also very short paths between the connected nodes despite the low density and relatively large size of the graphs.

*Auxiliary data statistics.* The MeSH data contained 719,877 terms and 54,935 tree codes, with ca. 2.371 tree code annotations per term in average. The statistics of the indices of related publications for SKIMMR and for gold standards are provided in Table 5. We provide values for the size of the index in numbers of publications covered ($|P|$) and an average number of related publications associated with each key ($\bar{R}$). The average length of the lists of related publications is much higher for all three instances of SKIMMR. This is a result of the small-world property of the SKIMMR networks which makes most of the publications connected with each other (although the connections mostly have weights close to zero).

### Evaluation results

In the following we report on the results measured using the specific SKIMMR knowledge bases and corresponding baseline data. Each category of the evaluation measures is covered

**Table 5** Statistics of the indices of related publications.

| Data set ID | Gold standard | | SKIMMR | |
|---|---|---|---|---|
| | $|P|$ | $\bar{R}$ | $|P|$ | $\bar{R}$ |
| SMA | 1,221 | 36.15 | 1,220 | 959.628 |
| PD | 4,727 | 28.61 | 4,724 | 4327.625 |
| TREC | 434 | 18.032 | 2,245 | 1251.424 |

in a separate section. Note that we mostly provide concise plots and summaries of the results here in the article, however, full results can be found online (Data Deposition).

*Semantic coherence.* Figure 5 shows the values of the aggregated semantic coherence measures (i.e., source-target coherence, product path coherence and average path coherence) for the PD, SMA and TREC data sets. The values were aggregated by computing their arithmetic means and are denoted by the *y*-axis of the plots. The *x*-axis corresponds to different combinations of the heuristics and path lengths for the execution of the random walks (as the coherence does not depend on the envelope size, this parameter is zero all the time in this case).[17] The combinations are grouped by heuristics (random preference, weight preference, similarity preference, dissimilarity preference from left to right). The path length parameter increases from left to right for each heuristic group on the *x*-axis. The green line is for the SKIMMR results and the blue line is for the PubMed baseline.

For any combination of the random walk execution parameters, SKIMMR outperforms the baseline by quite a large relative margin. The most successful heuristic in terms of coherence is the one that prefers more similar nodes to visit next (third quarter of the plots), and the coherence is generally lower for longer paths, which are all observations corresponding to intuitive assumptions.

*Information content.* Figure 6 shows the values of the arithmetic mean of all types of information content measures for the particular combinations of the random walk execution parameters (including also envelope sizes in increasing order for each heuristic). Although the relative difference is not as significant as in the semantic coherence case, SKIMMR again performs consistently better than the baseline. There are no significant differences between the specific heuristics. The information content increases with longer walks and larger envelopes, which is due to generally larger numbers of clusters occurring among more nodes involved in the measurement.

*Graph Structure.* Figure 7 shows the values of the clustering coefficient, again with green and blue lines for the SKIMMR and PubMed baseline results, respectively. SKIMMR exhibits larger level of complexity than the baseline in terms of clustering coefficient, with moderate relative margin in most cases. There are no significant differences between the particular walk heuristics. The complexity generally increases with the length of the path, but, interestingly enough, does not so with the size of the envelopes. The highest complexity is typically achieved for the longest paths without any envelope. We suspect

[17] The exact form of labels on the *x*-axis is a combination of heuristic (H), envelope diameter (E) and path length (L) parameters with their numeric identifiers (in case of heuristics) or values (for envelope size and path length). For instance, H = 2.E = 1.L = 10 stands for a measurement using the weight preference heuristic (identifier 2), envelope of diameter 1 and path of length 10.

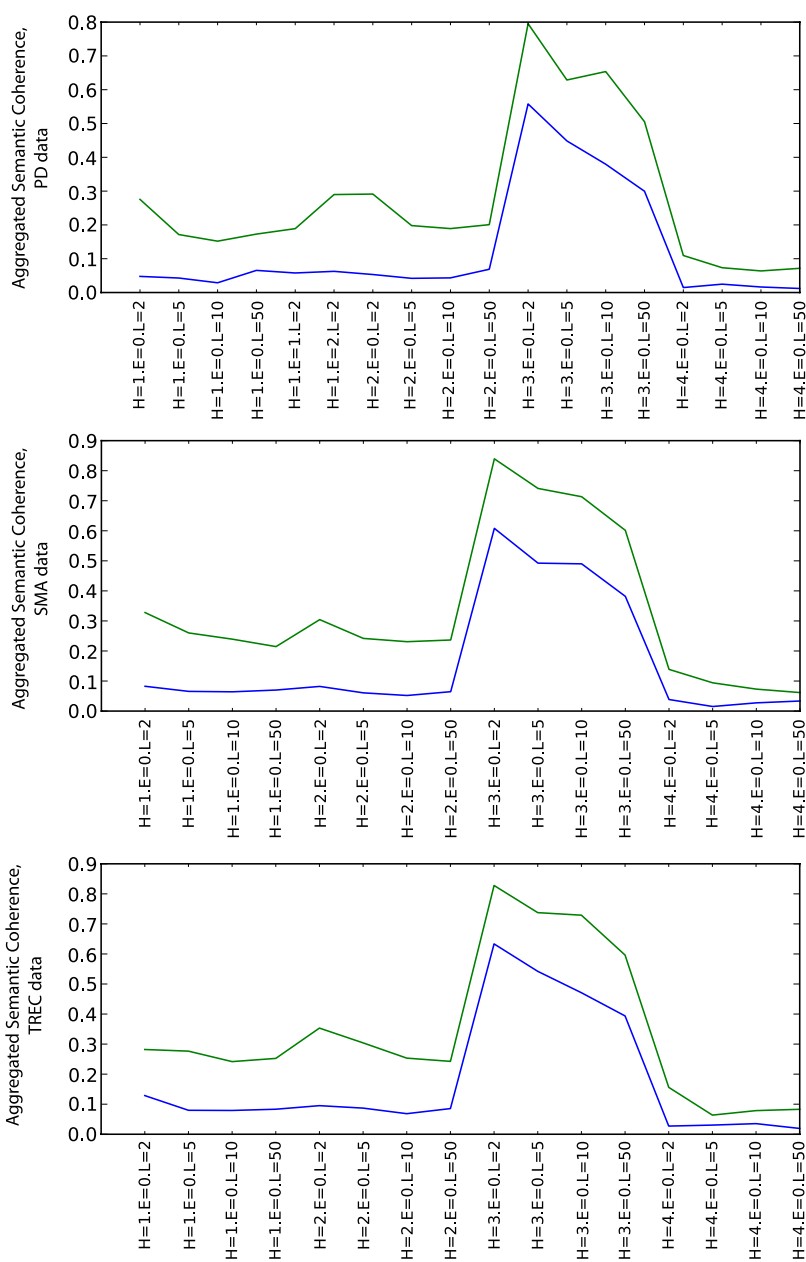

**Figure 5 Aggregated semantic coherence (blue: PubMed, green: SKIMMR).**

this to be related to the small world property of the graphs–adding more nodes from the envelope may not contribute to the actual complexity due to making the graph much more "uniformly" dense and therefore less complex.

*Auxiliary measures.* The number of clusters associated with the nodes on the paths (measures M and Q) is always higher for SKIMMR than for the PubMed baseline. The number of clusters associated with the whole envelopes (measures O and S) is almost always higher for SKIMMR with few exceptions of rather negligible relative differences in

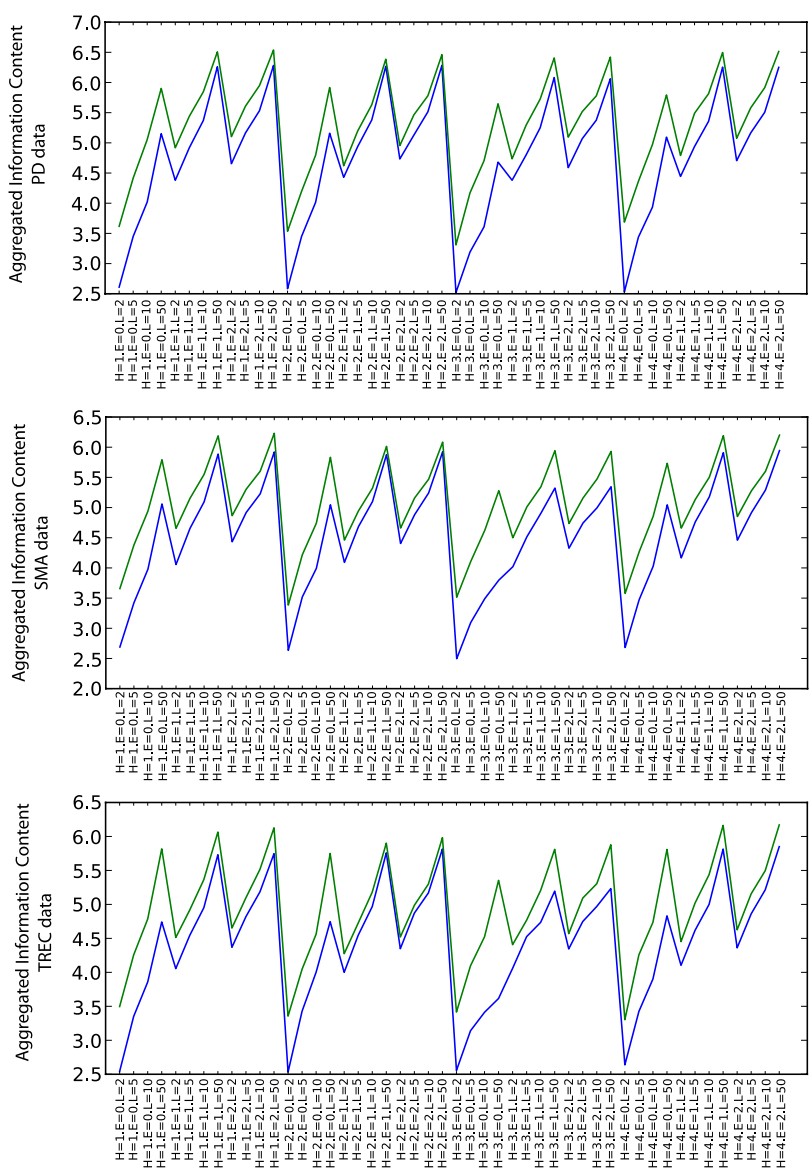

**Figure 6 Aggregated information content (blue: PubMed, green: SKIMMR).**

favour of the baseline. The average numbers of nodes per cluster on the path (measures N and R) are higher for SKIMMR except for the heuristic that prefers similar nodes to visit next. This can be explained by the increased likelihood of populating already "visited" clusters with this heuristic when traversing paths with lower numbers of clusters along them. Finally, the average number of nodes per cluster in the envelope (measures P and T) is higher for SKIMMR in most cases.

The general patterns observed among the auxiliary measure values indicates higher topical variability in the SKIMMR graphs, as there are more clusters that have generally higher cardinality than in the PubMed baselines. This is consistent with the observation of

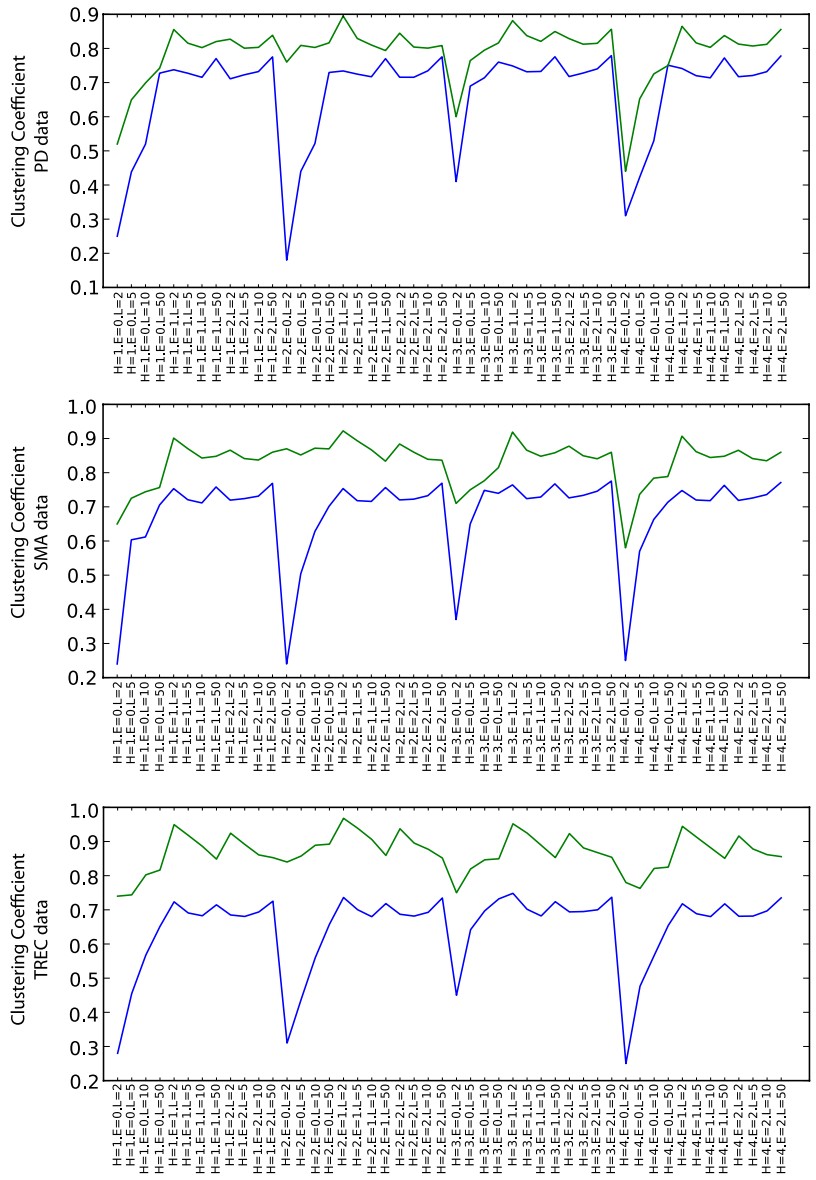

**Figure 7 Clustering coefficient (blue: PubMed, green: SKIMMR).**

the generally higher information content associated with the random walks in SKIMMR graphs.

*Related articles.* The results of the evaluation measures based on the lists of related articles generated by SKIMMR and by related baselines are summarised in Table 6. Note that as explained in 'Evaluation data', we used actual TREC evaluation data for the TREC dataset, while for PD and SMA, we used the related articles provided by PubMed due to negligible overlap with the TREC gold standard.

The $pre_{avg}$ and $rec_{avg}$ columns in Table 6 contain the precision and recall values for each data set, respectively, and the $C \geq 0.7$ contains the ratio of SKIMMR results that have

Nováček and Burns
2014
10.7717/peerj.483

**Table 6  Results for the related articles.**

| PD | | | SMA | | | TREC | | |
|---|---|---|---|---|---|---|---|---|
| $pre_{avg}$ | $rec_{avg}$ | $C \geq 0.7$ | $pre_{avg}$ | $rec_{avg}$ | $C \geq 0.7$ | $pre_{avg}$ | $rec_{avg}$ | $C \geq 0.7$ |
| 0.0095 | 0.0240 | 0.5576 | 0.0139 | 0.0777 | 0.5405 | 0.0154 | 0.0487 | 0.5862 |

significant correlation (i.e., at least 0.7) with the corresponding baseline. The absolute values of the average precision and recall are very poor, in units of percents. The correlation results are more promising, showing that more than half of the related document rankings produced by SKIMMR are reasonably aligned with the gold standard. Moreover, the correlation is highest for the TREC data set based on the only gold standard that is manually curated.

## DISCUSSION

SKIMMR provides a computational instantiation of the concept of 'skim reading.' In the early prototype stage, we generally focussed on delivering as much of the basic functionality as possible in a lightweight interface. Lacking enough representative data collected from ongoing user studies, we have designed a series of automated experiments to simulate several skim reading modes one can engage in with SKIMMR. We evaluated these experiments using gold standards derived from manually curated biomedical resources. Here we offer a discussion of the results in relation to the concept of machine-aided skim reading as realised by the SKIMMR prototype. The discussion is followed by an overview of related work and an outline of possible future directions.

### Interpreting the results

The secondary evaluation using lists of related publications induced by the SKIMMR knowledge bases did not bring particularly good results in terms of precision and recall. However, the correlation with the related document ranking provided by baselines was more satisfactory. This indicates that with better methods for pruning the rather extensive lists of related publications produced with SKIMMR, we may be able to improve the precision and recall substantially. Still, this evaluation was indirect since generating lists of related publications is not the main purpose of SKIMMR. Apart from indirect evaluation, we were also curious whether the data produced by SKIMMR could not be used also for a rather different task straightaway. The lesson learned is that this may be possible, however, some post-processing of the derived publication lists would be required to make the SKIMMR-based related document retrieval more accurate for practical applications.

Our main goal was to show that our approach to machine-aided skim reading can be efficient in navigating high-level conceptual structures derived from large numbers of publications. The results of the primary evaluation experiment—simulations of various types of skimming behaviour by random walks—demonstrated that our assumption may indeed be valid. The entity networks computed by SKIMMR are generally more *semantically coherent*, more *informative* and more *complex* than similar networks based on

the manually curated PubMed article annotations. This means that users will typically be able to browse the SKIMMR networks in a more focused way. At the same time, however, they will learn more interesting related information from the context of the browsing path, and can also potentially gain additional knowledge from more complex relationships between the concepts encountered on the way. This is very promising in the context of our original motivations for the presented research.

Experiments with actual users would have brought many more insights regarding the practical relevance of the SKIMMR prototype. Still, the simulations we have proposed cover four distinct classes of possible browsing behaviour, and our results are generally consistent regardless of the particular heuristic used. This leads us to believe that the evaluation measures computed on paths selected by human users would not be radically different from the patterns observed within our simulations.

## Related work

The text mining we use is similar to the techniques mentioned in *Yan et al. (2009)*, but we use a finer-grained notion of co-occurrence. Regarding biomedical text mining, tools like BioMedLEE (*Friedman et al., 2004*), MetaMap (*Aronson & Lang, 2010*) or SemRep (*Liu et al., 2012*) are closely related to our approach. The tools mostly focus on annotation of texts with concepts from standard biomedical vocabularies like UMLS which is very useful for many practical applications. However, it is relatively difficult to use the corresponding software modules within our tool due to complex dependencies and lack of simple APIs and/or batch scripts. The tools also lack the ability to identify concepts not present in the biomedical vocabularies or ontologies. Therefore we decided to use LingPipe's batch entity recogniser in SKIMMR. The tool is based on a relatively outdated GENIA corpus, but is very easy to integrate, efficient and capable of capturing unknown entities based on the underlying statistical model, which corresponds well to our goal of delivering a lightweight, extensible and easily portable tool for skim-reading.

The representation of the relationships between entities in texts is very close to the approach of *Baroni & Lenci (2010)*, however, we have extended the tensor-based representation to tackle a broader notion of text and data semantics, as described in detail in *Nováček, Handschuh & Decker (2011)*. The indexing and querying of the relationships between entities mentioned in the texts is based on fuzzy index structures, similarly to *Zadrozny & Nowacka (2009)*. We make use of the underlying distributional semantics representation, though, which captures more subtle features of the meaning of original texts.

Graph-based representations of natural language data have previously been generated using dependency parsing (*Ramakrishnan et al., 2008*; *Biemann et al., 2013*). Since these representations are derived directly from the parse structure, they are not necessarily tailored for the precise task of skim-reading but could provide a valuable intermediate representation. Another graph-based representation that is derived from the text of documents are similarity-based approaches derived from 'topic models' of document corpora (*Talley et al., 2011*). Although these analyses typically provide a visualization of the organization of documents, not of their contents, the topic modeling methods provide

statistical representation of the text that can then be leveraged to examine other aspects of the context of the document, such as its citations (*Foulds & Smyth, 2013*).

A broad research area of high relevance to the presented work is the field of 'Machine Reading' that can be defined as "*the autonomous understanding of text*" (*Etzioni, Banko & Cafarella, 2006*). It is an ambitious goal that has attracted much interest from NLP researchers (*Mulkar et al., 2007*; *Strassel et al., 2010*; *Poon & Domingos, 2010*). By framing the reading task as 'skimming' (which provides a little more structure than simply navigating a set of documents, but much less than a full representation of the semantics of documents), we hope to leverage machine reading principles into practical tools that can be used by domain experts straightforwardly.

Our approach shares some similarities with applications of spreading activation in information retrieval which are summarised for instance in the survey (*Crestani, 1997*). These approaches are based on associations between search results computed either off-line or based on the "live" user interactions. The network data representation used for the associations is quite close to SKIMMR, however, we do not use the spreading activation principle to actually retrieve the results. We let the users to navigate the graph by themselves which allows them to discover even niche and very domain-specific areas in the graph's structure that may not be reached using the spreading activation.

Works in literature based discovery using either semantic relationships (*Hristovski et al., 2006*) or corresponding graph structures (*Wilkowski et al., 2011*) are conceptually very similar to our approach to skim reading. However, the methods are quite specific when deployed, focusing predominantly on particular types of relationships and providing pre-defined schema for mining instances of the relationships from the textual data. We keep the process lightweight and easily portable, and leave the interpretation of the conceptual networks on the user. We do lose some accuracy by doing so, but the resulting framework is easily extensible and portable to a new domain within minutes, which provides for a broader coverage compensating the loss of accuracy.

From the user perspective, SKIMMR is quite closely related to GoPubMed (*Dietze et al., 2008*), a knowledge-based search engine for biomedical texts. GoPubMed uses Medical Subject Headings and Gene Ontology to speed up finding of relevant results by semantic annotation and classification of the search results. SKIMMR is oriented more on browsing than on searching, and the browsing is realised via knowledge bases inferred from the texts automatically in a bottom-up manner. This makes SKIMMR independent on any pre-defined ontology and lets users to combine their own domain knowledge with the data present in the article corpus.

Tools like DynaCat (*Pratt, 1997*) or QueryCat (*Pratt & Wasserman, 2000*) share the basic motivations with our work as they target the information overload problem in life sciences. They focus specifically on automated categorisation of user queries and the query results, aiming at increasing the precision of document retrieval. Our approach is different in that it focuses on letting users explore the content of the publications instead of the publications themselves. This provides an alternative solution to the information overload by leading

users to interesting information spanning across multiple documents that may not be grouped together by *Pratt (1997)* and *Pratt & Wasserman (2000)*.

Another related tool is Exhibit (*Huynh, Karger & Miller, 2007*), which can be used for faceted browsing of arbitrary datasets expressed in JSON (*Crockford, 2006*). Using Exhibit one can dynamically define the scope from which they want to explore the dataset and thus quickly focus on particular items of interest. However, Exhibit does not provide any solution on how to get the structured data to explore from possibly unstructured resources (such as texts).

Textpresso (*Müller, Kenny & Sternberg, 2004*) is quite similar to SKIMMR concerning searching for relations between concepts in particular chunks of text. However, the underlying ontologies and their instance sets have to be provided manually which often requires years of work, whereas SKIMMR operates without any such costly input. Moreover, the system's scale regarding the number of publications' full-texts and concepts covered is generally lower than the instances of SKIMMR that can be set up in minutes.

CORAAL (*Nováček et al., 2010*) is our previous work for cancer publication search, which extracts relations between entities from texts, based on the verb frames occurring in the sentences. The content is then exposed via a multiple-perspective search and browse interface. SKIMMR brings the following major improvements over CORAAL: (1) more advanced back-end (built using our distributional data semantics framework introduced in *Nováček, Handschuh & Decker, 2011*); (2) simplified modes of interaction with the data leading to increased usability and better user experience; (3) richer, more robust fuzzy querying; (4) general streamlining of the underlying technologies and front-ends motivated by the simple, yet powerful metaphor of machine-aided skim reading.

## Future work

Despite the initial promising results, there is still much to do in order to realise the full potential of SKIMMR as a machine-aided skim reading prototype. First of all, we need to continue our efforts in recruiting coherent and reliable sample user groups for each of the experimental SKIMMR instances in order to complement the presented evaluation by results of actual user studies. Once we get the users' feedback, we will analyse it and try to identify significant patterns emerging from the tracked behaviour data in order to correlate them with the explicit feedback, usability assessments and the results achieved in our simulation experiments. This will provide us with a sound basis for the next iteration of the SKIMMR prototype development, which will reflect more representative user requirements.

Regarding the SKIMMR development itself, the most important things to improve are as follows. We need to extract more types of relations than just co-occurrence and rather broadly defined similarity. One example of domain specific complex relation are associations of potential side effects with drugs. Another, more general example, is taxonomical relations (super-concept, sub-concept), which may help provide additional perspective to browsing the entity networks (i.e., starting with high-level overview of the relations between more abstract concepts and then focusing on the structure of the

connections between more specific sub-concepts of selected nodes). Other improvements related to the user interface are: (1) smoother navigation in the entity networks (the nodes have to be active and shift the focus of the displayed graph upon clicking on them, they may also display additional metadata, such as summaries of the associated source texts); (2) support of more expressive (conjunctive, disjunctive, etc.) search queries not only in the back-end, but also in the front-end, preferably with a dedicated graphical user interface that allows to formulate the queries easily even for lay users; (3) higher-level visualisation features such as evolution of selected concepts' neighbourhoods in time on a sliding scale. We believe that realisation of all these features will make SKIMMR a truly powerful tool for facilitating knowledge discovery (not only) in life sciences.

## ACKNOWLEDGEMENTS

We would like to thank to our former colleagues Eduard H. Hovy and Drashti Dave for their generously shared insights regarding the NLP and biomedical aspects, respectively, of the presented work. Last but not least, we are indebted to Maryann Martone for her guidance concerning the Spinal Muscular Atrophy domain and for multiple testing sessions during SKIMMR development which helped us to refine the tool in order to meet the actual requirements of life scientists.

## APPENDIX. FORMULAE DEFINITIONS

In this appendix we give full account on definitions of some of the formal notions used throughout the main article but not covered in detail there.

### Co-occurrences

The basic co-occurrence score $cooc((e_x, e_y), PubMed_{PMID})$ for two entities $e_x, e_y$ in an article $PubMed_{PMID}$, introduced in 'Extracting basic co-occurrence statements from texts', is computed as

$$cooc((e_x, e_y), PubMed_{PMID}) = \sum_{i,j \in S(e_x, e_y)} \frac{1}{1 + |i - j|} \qquad (1)$$

where $S(e_x, e_y)$ is a set of numbers of sentences that contain the entity $e_x$ or $e_y$ (assuming the sentences numbered sequentially from the beginning of the text). In practice, one may impose a limit on the maximum allowed distance of entities to be taken into account in the co-occurrence score computation (we disregard entities occurring more than 3 sentences apart from the score sum).

The non-normalised formula for corpus-wide co-occurrence for two outcomes (i.e., terms in our specific use case) $x, y$, using a base-2 logarithm (introduced in 'Corpus-wide co-occurrence'), is:

$$fpmi(x, y) = F(x, y) log_2 \frac{p(x, y)}{p(x)p(y)} \qquad (2)$$

where $F(x, y)$ is the absolute frequency of the $x, y$ co-occurrence and $p(x, y), p(x), p(y)$ are the joint and individual distributions, respectively. In our case, the distributions are the

weighted relative frequencies of the entity terms in the basic co-occurrence tuples generated from the input texts which are computed as follows. Let us assume a set $T$ of tuples

$$t_1 = (e_{1,x}, e_{1,y}, cooc((e_{1,x}, e_{1,y}), PubMed_{PMID_1}), PubMed_{PMID_1}),$$

$$t_2 = (e_{2,x}, e_{2,y}, cooc((e_{2,x}, e_{2,y}), PubMed_{PMID_2}), PubMed_{PMID_2}),$$

$$\vdots$$

$$t_n = (e_{n,x}, e_{n,y}, cooc((e_{n,x}, e_{n,y}), PubMed_{PMID_n}), PubMed_{PMID_n})$$

as a result of the basic co-occurrence statement extraction described in the previous section. The joint distribution of terms $x, y$ specific to our case can then be computed as:

$$p(x, y) = \frac{\sum_{w \in W(x,y,T)} w}{|T|} \tag{3}$$

where $W(x, y, T) = \{w | \exists e_1, e_2, w, i.(e_1, e_2, w, i) \in T \wedge ((e_1 = x \wedge e_2 = y) \vee (e_1 = y \wedge e_2 = x))\}$ is the set of weights in the basic co-occurrence tuples that contain both $x, y$ as entity arguments. Finally, the individual distribution of a term $z$ is computed as:

$$p(z) = \frac{\sum_{w \in W(z,T)} w}{|T|} \tag{4}$$

where $W(z, T) = \{w | \exists e_1, e_2, w, i.(e_1, e_2, w, i) \in T \wedge (e_1 = z \vee e_2 = z)\}$ is the set of weights in the basic co-occurrence tuples that contain $z$ as any one of the entity arguments. In the eventual result, all co-occurrence tuples with score lower than zero are omitted, while the remaining ones are normalised as follows:

$$npmi(x, y) = \nu(fpmi(x, y), P) \tag{5}$$

where $\nu$ is a function that divides the scores by the $P$-th percentile of all the scores and truncates the resulting value to 1 if it is higher than that. The motivation for such definition of the normalisation is that using the percentile, one can flexibly reduce the influence of possibly disproportional distributions in the scores (i.e., when there are few very high values, normalisation by the sum of all values or by the maximal value would result in most of the final scores being very low, whereas the carefully selected percentile can balance that out, reducing only relatively low number of very high scores to crisp 1).

## Similarities

Firstly we define the cosine similarity introduced in 'Similarity'. For that we need few auxiliary notions. First of them is a so called 'co-occurrence complement' $\bar{x}$ of an entity $x$:

$$\bar{x} = \{(e, w) | \exists e, w.(e, cooc, x, w) \in KB \vee (x, cooc, e, w) \in KB\} \tag{6}$$

where $KB$ is the knowledge base, i.e., the set of the aggregated co-occurrence statements computed as shown in 'Corpus-wide co-occurrence'. Additionally, we define an element-set projection of an entity's co-occurrence complement $\bar{x}$ as $\bar{x}_1 = \{y | \exists w.w \neq 0 \wedge (y, w) \in \bar{x}\}$,

i.e., set of all the entities in the co-occurrence complement abstracting from the corresponding co-occurrence weights. Finally, we use a shorthand notation $\bar{x}[y] = w$ such that $(y, w) \in \bar{x}$ for a quick reference to the weight corresponding to an entity in a co-occurrence complement. If an entity $y$ is missing in the co-occurrence complement of $x$, we define $\bar{x}[y] = 0$.

**Example 8** *Assuming that the knowledge base consists only from one co-occurrence tuple* $(\texttt{parkinsonism}, \texttt{cooc}, \texttt{DRD}, 0.545)$ *from the previous Example 2, we can define two co-occurrence complements on the entities in it:*

$$\overline{\texttt{parkinsonism}} = \{(\texttt{DRD}, 0.545)\}, \qquad \overline{\texttt{DRD}} = \{(\texttt{parkinsonism}, 0.545)\}.$$

*The element-set projection of* $\overline{\texttt{parkinsonism}}$ *is then a set* $\{\texttt{DRD}\}$*, while* $\overline{\texttt{parkinsonism}}[\texttt{DRD}]$ *equals* $0.545$*.*

Now we can define the similarity between two entities $a, b$ in a SKIMMR knowledge base as:

$$sim(a, b) = \frac{\sum_{z \in \bar{a}_1 \cap \bar{b}_1} \bar{a}[z]\bar{b}[z]}{\sqrt{\sum_{x \in \bar{a}_1} \bar{a}[x]^2} \sqrt{\sum_{y \in \bar{b}_1} \bar{b}[y]^2}} \tag{7}$$

where $\bar{a}, \bar{b}$ are the co-occurrence complements of $a, b$, and $\bar{a}_1, \bar{b}_1$ their element-set projections. It can be easily seen that the formula directly corresponds to the definition of cosine distance: its top part is the dot product of the co-occurrence context vectors corresponding to the entities $a, b$, while the lower part is multiplication of the vectors' sizes (Euclidean norms in particular).

The MeSH-based semantic similarity of entities, introduced in 'Running and measuring the random walks', is defined as

$$sim_M(X, Y) = \max_{u \in C_S(X), v \in C_S(Y)} \frac{2 \cdot dpt(lcs(u, v))}{dpt(u) + dpt(v)} \tag{8}$$

where the specific tree codes in the $C_S(X), C_S(Y)$ are interpreted as nodes in the MeSH taxonomy, the *lcs* stands for the least common subsumer of two nodes in the taxonomy and *dpt* is the depth of a node in the taxonomy (defined as zero if no node is supplied as an argument, i.e., if *lcs* has no result). The formula we use is essentially based on a frequently used taxonomy-based similarity measure defined in *Wu & Palmer (1994)*. We only maximise it across all possible cluster annotations of the two input entities to find the best match. Note that this strategy is safe in case of a resource with as low ambiguity as MeSH – while there are often more annotations of a term, they do not refer to different senses but rather to different branches in the taxonomy. Therefore using the maximum similarity corresponds to finding the most appropriate branch in the MeSH taxonomy along which the terms can be compared.

## Entropies

'Running and measuring the random walks' introduced entropies for expressing information value of SKIMMR evaluation samples (*i.e.*, random walks and their contexts). The entropies are defined using the notion of MeSH cluster size ($cs(\ldots)$) introduced in the main part of the article. Given a set $Z$ of nodes of interest, the entropy based on MeSH cluster annotations, $H_M(Z)$, is computed as

$$H_M(Z) = -\sum_{C_i \in C(Z)} \frac{cs(C_i)}{\sum_{C_j \in C(Z)} cs(C_j)} \cdot \log_2 \frac{cs(C_i)}{\sum_{C_j \in C(Z)} cs(C_j)} \tag{9}$$

where $C$ is one of $C_A, C_S$, depending whether we consider the abstract or the specific nodes. Similarly, the component-based entropy $H_C(Z)$ is defined as

$$H_C(Z) = -\sum_{C_i \in B(Z)} \frac{|C_i|}{\sum_{C_j \in B(Z)} |C_j|} \cdot \log_2 \frac{|C_i|}{\sum_{C_j \in B(Z)} |C_j|} \tag{10}$$

where $B(Z)$ is a function returning a set of biconnected components in the envelope $Z$, which is effectively a set of subsets of nodes from $Z$.

## Precision and recall

The indices of related articles are compared using precision and recall measures, as stated in 'Comparing the indices of related articles'. Let $I_S : P \to 2^P, I_G : P \to 2^P$ be the SKIMMR and gold standard indices of related publications, respectively ($P$ being a set of publication identifiers). Then the precision and recall for a publication $p \in P$ are computed as

$$pre(p) = \frac{|I_S(p) \cap I_G(p)|}{|I_S(p)|}, \qquad rec(p) = \frac{|I_S(p) \cap I_G(p)|}{|I_G(p)|} \tag{11}$$

respectively. To balance the possibly quite different lengths of the lists of related articles, we limit the computation of the precision and recall up to at most 50 most relevant items in the lists. The average values of precision and recall for a corpus of articles $X \subseteq P$ are computed as

$$pre_{avg}(X) = \frac{\sum_{p \in X} pre(p)}{|X|}, \qquad rec_{avg}(X) = \frac{\sum_{p \in X} rec(p)}{|X|} \tag{12}$$

respectively.

### Funding

This publication has emanated from research supported in part by research grants from Science Foundation Ireland (SFI) under Grant Numbers SFI/08/CE/I1380, SFI/08/CE/I1380 – STTF 11 (2), and SFI/12/RC/2289. Work was also supported under NIH grants RO1-GM083871 and RO1-MH079068-01A2. The funders had no role in study design, data collection and analysis, decision to publish, or preparation of the manuscript.

## Grant Disclosures

The following grant information was disclosed by the authors:
Science Foundation Ireland (SFI): SFI/08/CE/I1380, SFI/08/CE/I1380–STTF 11 (2), SFI/12/RC/2289.
NIH: RO1-GM083871, RO1-MH079068-01A2.

## Competing Interests

The authors declare there are no competing interests.

## Author Contributions

- Vít Nováček conceived and designed the experiments, performed the experiments, analyzed the data, contributed reagents/materials/analysis tools, wrote the paper, prepared figures and/or tables, reviewed drafts of the paper, implemented the SKIMMR prototype and corresponding experimental validation scripts.
- Gully A.P.C. Burns conceived and designed the experiments, wrote the paper, reviewed drafts of the paper.

## Ethics

The following information was supplied relating to ethical approvals (i.e., approving body and any reference numbers):
IRB name: USC UPIRB, approval number: UP-12-00414.

## Data Deposition

The following information was supplied regarding the deposition of related data:
http://skimmr.org/resources/skimmr/pd.tgz (Parskinson's Disease instance of SKIMMR, canned archive version)
http://skimmr.org/resources/skimmr/sma.tgz (Spinal Muscular Atrophy instance of SKIMMR, canned archive version)
http://skimmr.org/resources/skimmr/trec.tgz (TREC instance of SKIMMR, canned archive version)
http://skimmr.org/resources/skimmr/plots.tgz (complete plots of the results).

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
