# Peer review of "SKIMMR: facilitating knowledge discovery in life sciences by machine-aided skim reading"

_PeerJ, doi:10.7717/peerj.483_

## Round 0.1 · original submission · Major Revisions

· Academic Editor

Major Revisions

The reviewers have identified several significant concerns that should be addressed in a revision. As I read the reviews, the following items seem to me to be the most critical:

1. Background discussions omit important prior work.
2. Inaccuracies in discussion of Pubmed and some biomedical ontologies.
3. Lack of detail on concept extraction.
4. Lack of evaluation. Although the paper describes ongoing user tests, no results are provided.

Perhaps most critically, this paper does not present a complete evaluation. The evaluation framework presented in the paper fail to meet the requirements for experimental design and validity of findings described in the PeerJ editorial criteria (https://peerj.com/about/editorial-criteria/). A revised version should addresss these concerns.

·

Basic reporting

The manuscript is very nicely written using clear and understandable English with very few spelling or grammatical errors.

The Introduction and related work sections do demonstrate how the work fits into the broader field of knowledge. Also, the sections cite literature appropriately but are missing some important related systems [1-3]. I have more to say on this in the discussion of the validity of findings. Some statements and points could be presented more accurately. One is that on line 55 the authors imply that PubMed ranks its search results. This is not true since PubMed uses Boolean retrieval. The initial results are *sorted* by date of indexing within PubMed. Users have many other options for sorting. PubMed Related Article Search (see [4]) does rank by document "eliteness" but this is not the default search interface. In the relate work section, the authors correctly compare their work to GoPubMed but do not do a good enough job of describing the system (e.g., that it shows users text snippets along with tags from ontologies) or the ontologies that are used. GoPubMed uses Medical Subject Headings (MeSH) and the Gene Ontology (GO) both of which cannot simply be dismissed as "rigid, manually built ontologies". Rather, they are quite comprehensive for the biomedical and biological domains respectively, are the product of many thousands of man hours of collaborative development among domain experts, and their benefits for information retrieval have been well studied.

Figure seem relevant and at an acceptable resolution. Some of the tables shown in the examples contain font sizes that are too small.

1. W. Pratt and H. Wasserman. Querycat: Automatic categorization of medline queries, 2000.


2. Wanda Pratt. Dynamic organization of search results using the UMLS. In AMIA Proceeding Fall 1997, pages 480-484, 1997.

3. http://www-01.ibm.com/software/data/information-optimization/

4. Jimmy Lin and W John Wilbur, “PubMed related articles: a probabilistic topic-based model for content similarity,” BMC Bioinformatics 8, no. 1 (2007): 423.

Experimental design

Its not clear to me that the work described fits with the scope of the journal because an actual experiment is not being reported. The manuscript reports in very nice detail the design of a novel information retrieval system, providing several examples to motivate the methods discussion. The authors also suggest an experimental design and point to tooling on the system's website to support the proposed experiment. However, I am not sure if these laudible efforts satisfy the scope of the journal. For example, the system design might be better placed in a journal like the Journal for Biomedical Informatics or IEEE Transactions on Information Technology in Biomedicine [2]. Likewise, the experimental design might be better placed in the JMIR Research Protocols [3]. Apart from these significant concerns, the methods writeup and the downloadable/accessible software artifacts are all good quality. Issues that could easily addressed include 1) the authors should make clear what the colored edges represent in the graph figures, the use of PURLs for URIs to the project artifacts would make the links easier to use and more permanent, No ethical concerns.

1. www.journals.elsevier.com/journal-of-biomedical-informatics/‎
2. http://ieeexplore.ieee.org/xpl/RecentIssue.jsp?reload=true&punumber=4233
3. www.researchprotocols.org/‎

Validity of the findings

I have no concerns about the actual software artifacts or the computational methods reported. The biggest concerns have to do with the conclusions. The authors claim that the importance of their work is that it contributes a novel paradigm for search, skimming. However, the proposed methods seems to me to be quite similar to "spreading activation search" (aka "associative retrieval") discussed in [1-3]. Thus, the novelty is questionable. Important aspect of the target documents (PubMed abstracts) are not discussed such as that all PubMed abstracts are already tagged with topical concepts from the Medical Subject Headings vocabulary. It would seem that this high-quality, curated, data would be very valuable to the use case. Another serious concern is the lack of detail provided on how concepts are extracted from documents. All that is provided is a like to the LingPipe toolkit. The obvious question is how accurate the extaraction approach is. After some searching of my own, I found that, for the biomedical domain, LingPipe uses HMM classifiers based on the GENIA corpus. This is an old corpus and one question is how useful and valid the extraction is compared to modern tools such as the UMLS MetaMap and SemRep tools [4] or the Bioportal Annotator which uses MGrep and a few hundred ontologies [5]. In the discussion, the comments on the difficulty of acquiring participants for user studies seem anecdotal and detract from the paper. Also, I question why a more conventional IR evaluation could not be designed using existing corpora such as that developed for the TREC Genomics Track [6] that was used in evaluation of both GoPubMed and PubMed Related Article Search.

1. R. K. Belew. Adaptive information retrieval: using a connectionist representation to retrieve and learn about documents. In SIGIR '89: Proceedings of the 12th annual international ACM SIGIR conference on Research and development in information retrieval, pages 11-20, New York, NY, USA, 1989. ACM Press.

2. Richard K. Belew. Finding out about: a cognitive perspective on search engine technology and the WWW. Cambridge, 2000.

3. F. Crestani. Application of spreading activation techniques in informationretrieval. Artif. Intell. Rev., 11(6):453-482, 1997.

4. http://www.ncbi.nlm.nih.gov/pmc/articles/PMC3540517/

5. Noy et al. BioPortal: Ontologies and integrated data resources at the click of a mouse. Nucleic Acids Research 37:W170-173, 2009

6. Cohen AM, Hersh WR. The TREC 2004 genomics track categorization task: classifying full text biomedical documents. J Biomed Discov Collab. 2006 Mar 14;1:4. PubMed PMID: 16722582; PubMed Central PMCID: PMC1440303

Reviewer 2 ·

Basic reporting

The writing style, structure, and figures in the sections through 3.2 are good. Starting with 3.3, the text begins to sound like an advertisement for the software package and any content or analysis is minimal. Although the literature review covers development literature on similar systems, the literature on reading behavior/reading patterns is missing. For example, Tenopir & King, Vakkari, Nicholas and Rowlands.

Experimental design

The problem for which this system was developed is clearly defined, although hypotheses should be more explicitly stated. This is an important issue, how have you tested that your system solves the human problem described in the introduction? The development and testing of the system with documents appears to be conducted with rigour and to a high standard. Then something puzzling occurs—the authors speak about methods for ongoing user studies and evaluation. Results are mostly absent, however and we are left with methods, but no corresponding analysis. It is at this point that the reader begins to lose confidence in what had been presented well until this point. It seems that internal testing of the algorithms occurred during software development, but planned testing of the software with human subjects didn’t happen or didn’t succeed. It becomes unclear.

Validity of the findings

Again, the system design, algorithms, and description are interesting. For user tests that were proposed the results are missing. Conclusions seem to be at a software sales level, rather than a rigorous test level. Will the code be made available for replication or further testing? Should this paper wait to be published until the user tests that are described get done?

Additional comments

This paper starts out in a very promising way. Introduction to the problem, discussion of your system and approach are all interesting and detailed. Then, starting at section 3.3 it falls apart. Analysis and findings start sounding like a software advertisement, user studies are alluded to but findings are absent, and it just peters out.

---

## Round 0.2 · Minor Revisions

· Academic Editor

Minor Revisions

Reviewer 1 has made several suggestions regarding the organization of the paper (Basic reporting); formatting - particularly with regards to LaTeX (Validity of the findings) and comments for the author. Comments under "Validity of the findings" raise some points that might be addressed in the discussion.

Addressing these concerns should not be terribly difficult. The resulting revision is likely to be substantially stronger.

·

Basic reporting

I applaud the effort that the authors have taken to address the first set of critiques. The addition of an evaluation is a great improvement. Also, the review of related work is much more comprehensive. Unfortunately, the structure of the article is now a bit tangled with the addition of the new evaluations. As it is, the article is cumbersome to read and far too technically detailed for the reader to get the important points and adequately assess the validity of results. The methods for the evaluation should be placed in the "methods" section rather than the results. There is now a great excess of formulas and other technical information that would be best placed in a supplement. Noted limitations (e.g, the use of the GENIA corpus) would be better placed in a "Potential Lmitations" section. For readibility, I think that all formulas should be numbered and many of them should go into the supplement since they seem pretty typical for the domain.

Experimental design

The simulations are intriguing, creative, and reportable. The results do not seem overstated and its clear that further work requires a solid user study which is a stated goal of future work.

Validity of the findings

The results seem to have face validity and this reviewer can find no obvious source of bias that would limit the author's finding beyond what they have stated. The precision and recall analysis yielded apparently poor results and deriving these statistics from overlap of related articles requires some more thought. This is because the original algorithm for PubMed related articles is not really a gold standard because it is known to produce false associations ( see its "precision at 5" in the original evaluation). So, its just a comparative algorithm which might suggest that using agreement over chance (e.g., Kappa) might be a better metric than Prec and Recall and more informative than correlation. That being said, this reviewer did not have time to work through the numerous formulas to check for proper usage and accuracy. It seems that numerous equations have a dangling or non-relevant apostrophe which might be a side-effect of the use of commas at the end of formulas written in LaTeX. For example, the first formula in section 2.1:

$$
cooc((e_x,e_y),PubMed_{PMID})=\sum_{i,j \in S(e_x,e_y)} \frac{1}{1+|i - j|},
$$

This seems to be a very common problem in the paper.

Additional comments

Minor comments:
Abstract - 2nd sentence of results - missing 'the': 'that are <the> most relevant'

Abstract - I think that "formal comparison" should be broken into the methods and results. Describing a methodology in the results is not typical. The authors did an evaluation so report it as a scientific evaluations

2.2.1 third sentence - co-occuring terms <are> associated

Footnotes: the use of raw IP addresses to link to materials is not robust over time. Suggest changing to PURLS before publication

line 390 - wrong single quote symbol before 'Magnetic

p22, par 2: 'we construct *an* edge' (currently 'and')

near line 460: 'we use either <an> index...'

Figures - the y axes are bit hard to see and the X axis nearly impossible.

Reviewer 2 ·

Basic reporting

see below

Experimental design

see below

Validity of the findings

see below

Additional comments

The authors have addressed all of the issues brought up by the reviewers. Of most concern and most importantly, they have devised an empirical test which instills more confidence in their conclusions. Although the test is based on anticipated behavior, I am satisfied that it provides evidence of the system performance to back up some of the claims. In addition, they have substantially rewritten the literature review, grounding their work in the context of other similar work. I therefore recommend publication of this revised version.

---

## Round 0.3 · accepted · Accept

· Academic Editor

Accept

Thank you for addressing the concerns raised by reviewer 1 - these revisions have clearly made the paper stronger.